# Molecular Surveillance of Artemisinin-Based Combination Therapies Resistance in *Plasmodium falciparum* Parasites from Bioko Island, Equatorial Guinea

YaQun Liu,[a] XueYan Liang,[a] ⓘJian Li,[b] JiangTao Chen,[c,d] HuiYing Huang,[a,b] YuZhong Zheng,[a] JinQuan He,[c] Carlos Salas Ehapo,[e] Urbano Monsuy Eyi,[e] PeiKui Yang,[a] LiYun Lin,[a] WeiZhong Chen,[f] GuangYu Sun,[f] XiangZhi Liu,[f] GuangCai Zha,[a] JunLi Wang,[g] ChunFang Wang,[g] HuaGui Wei,[g] ⓘMin Lin[a,g]

[a]School of Food Engineering and Biotechnology, Hanshan Normal University, Chaozhou, Guangdong, People's Republic of China
[b]School of Basic Medical Sciences, Hubei University of Medicine, Shiyan, Hubei, People's Republic of China
[c]The Chinese Medical Aid Team to the Republic of Equatorial Guinea, Guangzhou, Guangdong, People's Republic of China
[d]Department of Medical Laboratory, Huizhou Central Hospital, Huizhou, Guangdong, People's Republic of China
[e]Department of Medical Laboratory, Malabo Regional Hospital, Malabo, Equatorial Guinea
[f]Department of Medical Laboratory, Chaozhou People's Hospital Affiliated to Shantou University Medical College, Chaozhou, Guangdong, People's Republic of China
[g]School of Laboratory Medicine, Youjiang Medical University for Nationalities, Baise, Guangxi, People's Republic of China

YaQun Liu and XueYan Liang contributed equally to this article. Author order was determined by the corresponding author after negotiation.

**ABSTRACT** Artemisinin-based combination therapies (ACTs) resistance has emerged and could be diffusing in Africa. As an offshore island on the African continent, the island of Bioko in Equatorial Guinea is considered severely affected and resistant to drug-resistant *Plasmodium falciparum* malaria. However, the spatial and temporal distribution remain unclear. Molecular monitoring targeting the *Pfcrt*, *Pfk13*, *Pfpm2*, and *Pfmdr1* genes was conducted to provide insight into the impact of current antimalarial drug resistance on the island. Furthermore, polymorphic characteristics, haplotype network, and the effect of natural selection of the *Pfk13* gene were evaluated. A total of 152 *Plasmodium falciparum* samples (collected from 2017 to 2019) were analyzed for copy number variation of the *Pfpm2* gene and *Pfk13*, *Pfcrt*, and *Pfmdr1* mutations. Statistical analysis of *Pfk13* sequences was performed following different evolutionary models using 96 Bioko sequences and 1322 global sequences. The results showed that the prevalence of *Pfk13, Pfcrt,* and *Pfmdr1* mutations was 73.68%, 78.29%, and 75.66%, respectively. Large proportions of isolates with multiple copies of *Pfpm2* were observed (67.86%). In Bioko parasites, the genetic diversity of *Pfk13* was low, and purifying selection was suggested by Tajima's D test ($-1.644$, $P > 0.05$) and the dN/dS test ($-0.0004438$, $P > 0.05$). The extended haplotype homozygosity analysis revealed that *Pfk13*_K189T, although most frequent in Africa, has not yet conferred a selective advantage for parasitic survival. The results suggested that the implementation of continuous drug monitoring on Bioko Island is an essential measure.

**IMPORTANCE** Malaria, one of the tropical parasitic diseases with a high transmission rate in Bioko Island, Equatorial Guinea, especially caused by *P. falciparum* is highly prevalent in this region and is commonly treated locally with ACTs. The declining antimalarial susceptibility of artemisinin-based drugs suggested that resistance to artemisinin and its derivatives is developing in *P. falciparum*. Copy number variants in *Pfpm2* and genetic polymorphisms in *Pfk13*, *Pfcrt*, and *Pfmdr1* can be used as risk assessment indicators to track the development and spread of drug resistance. This study reported for the first time the molecular surveillance of *Pfpm2*, *Pfcrt*, *Pfk13*, and *Pfmdr1* genes in Bioko Island from 2017 to 2019 to assess the possible risk of local drug-resistant *P. falciparum*.

**Ad Hoc Peer Reviewer** Mateuz Placinski, ⓘOriana Kreutzfeld, University of California, San Francisco

Address correspondence to Jian Li, yxlijian@163.com, or Min Lin, konfutea@hotmail.com.

The authors declare no conflict of interest.

**KEYWORDS** malaria, artemisinin combination therapies, drug resistance, mutation, natural selection, Equatorial Guinea

**M**alaria is one of the most important tropical parasitic diseases, with an estimated 241 million cases and 627,000 deaths in 2020 (1). The emergence of *Plasmodium falciparum* (*P. falciparum*) resistance to antimalarial drugs has been threatening the global control of malaria and elimination efforts (2, 3). Currently, the World Health Organization (WHO) has recommended artemisinin-based combination therapies (ACTs) as a first-line treatment since 2001, which has greatly contributed to the reduction in malaria morbidity and mortality (4). One of the rationales for ACTs use is to maintain the long-term efficacy of artemisinins against *P. falciparum*. Six ACTs are currently recommended by the WHO for treating malaria cases worldwide: (i) artesunate-amodiaquine (AS/AQ), (ii) artemether-lumefantrine (AL), (iii) artesunate-sulfadoxine-pyrimethamine (AS/SP), (iv) artesunate-mefloquine (AS/MQ), (v) artesunate-pyronaridine (AS/PY) and (vi) dihydroartemisinin-piperaquine (DHA/PPQ) (5–7).

To achieve this objective, the choice of partner drugs of artemisinins is important and should be made based on their effectiveness on circulating local parasite isolates. Consequently, it is essential to monitor drug-resistant molecular markers of ACTs in different geographical and areas of endemicity, which will facilitate the selection of partner drugs associated with artemisinins in ACTs. In addition, assessing the genetic diversity and changes in the efficacy of natural selection of *P. falciparum* populations under ACTs pressure in a different area may allow us to track the emergence of drug-resistant mutations and malaria parasites circulating across different geographic areas (8).

Molecular markers of drug resistance are useful for identifying drug-resistant *P. falciparum*. Artemisinin resistance, defined as delayed parasite clearance (9), is associated with mutations in the propeller domain of the *P. falciparum* kelch13 (*Pfk13*) gene (PF3D7_1343700) (10). Previous studies have confirmed that mutation K76T in the *Pfcrt* gene can reduce the parasite's susceptibility to antimalarial drugs, such as chloroquine, quinine, AQ, and PPQ (11, 12). Resistance to piperaquine has been linked to increased copy number variations (CNVs) of the *P. falciparum* plasmepsin II (*Pfpm2*) gene (PF3D7_1408000) (13). Mutations in *P. falciparum* multidrug resistance protein 1 (*Pfmdr1*) have been associated with various parasite responses to amodiaquine and lumefantrine (14). *In vitro* experiments have shown that the N86Y mutation (*Pfmdr1*_N86Y) could increase the 50% inhibitory concentration ($IC_{50}$) of amodiaquine (15). Reduced susceptibility to lumefantrine has been associated with the *Pfmdr1*_Y184F mutation (16).

As an island off the coast of the African continent, Bioko Island, with historically high malaria transmission in Equatorial Guinea, has been subject to extensive interventions, including intensive vector control, improved case management, intermittent preventative treatments (IPT), and behavioral change interventions under the Bioko Island Malaria Control Project (BIMCP) (2004 to 2018) and now under the Bioko Island Malaria Elimination Project (BIMEP) (2019 to 2025) (17). These supplies included the introduction of indoor residual spraying (IRS) and the distribution of insecticide-treated nets (ITNs) to all households on Bioko Island, as well as free first-line ACTs, which reduced parasite prevalence from 45.0% to 8.5% between 2004 and 2016. ACTs, including AL, AS/AQ, AS/SP, and DHA/PPQ have successfully replaced chloroquine as the first-line antimalarial drug on Bioko Island, Equatorial Guinea (https://mcdinternational.org/index). The use of oral artemisinin monotherapy was forbidden in Equatorial Guinea in 2014. Although ACTs are available and free, most individuals still sought their medicines in private pharmacies rather than receiving treatment from public health services. Bioko Island reported falsified ACTs with prevalence rates ranging between 6.1% and 16.1%, depending on the sampling method used (18). All the above-mentioned issues increase the risk of drug resistance on the island. It should be noted that to date, there have been no reports on the efficacy of DHA/PPQ on Bioko Island.

In this study, to evaluate the resistance of *P. falciparum* to antimalarial drugs on Bioko Island of Equatorial Guinea, molecular surveillance targeting the *Pfcrt*, *Pfk13*,

**TABLE 1** Profiles of the mutations in *Pfmdr1*, *Pfcrt*, and *Pfk13* genes

| Gene | Codon position | Nucleotide reference | Nucleotide mutation[a] | Polymorphism[a,b] | MF (%)[c] |
|---|---|---|---|---|---|
| *Pfmdr1* | 86 | AAT | **T**AT | N86**Y** | 22.69% |
|  | 184 | TAT | T**T**T | Y184**F** | 43.70% |
| *Pfcrt* | 74-76 | ATG AAT AAA | AT**T GAA A**CA | CVMNK72-76CV**IET** | 26.95% |
| *Pfk13* | 136 | CAT | **A**AT | H136**N** | 1.04% |
|  | 189 | AAA | AA**C** | K189**N** | 2.08% |
|  | 189 | AAA | A**C**A | K189**T** | 53.12% |
|  | 193-194 | GTA AAT | GT/**CA**/**G G**AT | VN193-194V/**AD** | 1.04% |
|  | 213 | AGT | **G**/AG/**C**T | S213**G/A/T** | 1.04% |

[a]Mutations are in boldface.
[b]*Pfk13* VN193-194V/AD and *Pfk13* S213G/A/T were mixed type mutations.
[c]MF, mutation frequency.

*Pfpm2*, and *Pfmdr* genes was conducted. The polymorphic characteristics, haplotype network, and effect of natural selection of the *Pfk13* gene were evaluated. This might contribute to the subsequent treatment of malaria patients and malaria control.

## RESULTS

**Mutations of *Pfk13*, *Pfcrt* and *Pfmdr1* in Bioko isolates.** From January 2017 to December 2019, we collected samples from 152 *P. falciparum*-infected patients on Bioko Island, Equatorial Guinea. The median (interquartile range [IQR]) age of patients was 26 years (18 to 38.5), and 36.5% of them were women. The median (IQR) parasite density was 3272 parasites/$\mu$L (1527 to 7940).

There were 119 and 115 *P. falciparum* isolates from the cases successfully sequenced for the *Pfmdr1* (78.29%, 119/152) and *Pfcrt* (75.66%, 115/152) loci, respectively. Allelic variation in *Pfmdr1* was observed only at codons 86 and 184. The prevalence of *Pfmdr1*_N86Y and *Pfmdr1*_Y184F mutations was 22.69% and 43.70%, respectively. Two haplotypes, CVMNK (wild type) and triple mutant allele CVIET (mutation type) were present at *Pfcrt* codons 72 to 76 with a mutation prevalence of 26.95%. A total of 112 (73.68%, 112/152) *Pfk13* sequences were successfully amplified in the samples, and only 96 of them were available for subsequent analysis due to their long sequences and poor partial sequencing results. Using PF3D7_1343700 as the reference sequence, five nonsynonymous mutations (H136N, K189N, K189T, VN193-194V/AD, and S213G/A/T) were observed, but none of them was associated with artemisinin delayed parasite clearance (19–21). Among them, *Pfk13*_K189T (53.12%) was the most prevalent in the Bioko isolates, while the other loci were mutated at a lower frequency. The mutations at codons 193, 194, and 213 were mixed type, with amino acids mutated from VN to V/AD and S to G/A/T, respectively (Table 1).

**Pfpm2 gene copy number variation in Bioko isolates.** A total of 56 (36.60%, 56/152) out of 152 samples successfully amplified the *Pfpm2* gene. By using a copy number threshold of 1.5 to define single copy or multiple copies, 30.36% (0.721 to 1.394, 17/56) of the isolates showed a single copy of *Pfpm2* gene amplification, while 67.86% (1.516 to 2.889, 38/56) of the isolates carried multiple copies of *Pfpm2*. The 3D7 *P. falciparum* has a single copy of the *Pfmd2* gene with CNVs ranging from 1.006 to 1.122 (Fig. 1). Based on the increased CNVs of the *Pfpm2* gene on Bioko Island compared to the standard strain 3D7 of *P. falciparum*, it is hypothesized that the *Pfpm2* CNVs may serve as a risk assessment indicator for tracking the development and spread of DHA/PPQ drug resistance on the island.

**Genetic diversity and effect of natural selection of the Bioko Pfk13 gene.** The nucleotide diversity and natural selection were analyzed in the 96 Bioko *Pfk13* sequences. The average number of pairwise nucleotide differences (K) was 0.699. The highest number of nucleotide differences (K = 0.637) was detected in the *Plasmodium*-specific region (codons 1 to 440), while the fewest (K = 0.062) were found in the propeller domain (codons 441 to 726). The overall haplotype diversity (Hd) and nucleotide diversity ($\pi$) for this gene were estimated to be 0.596 $\pm$ 0.030 and 0.00046, respectively. Estimation of the number of synonymous substitutions per site (dS) and the number of

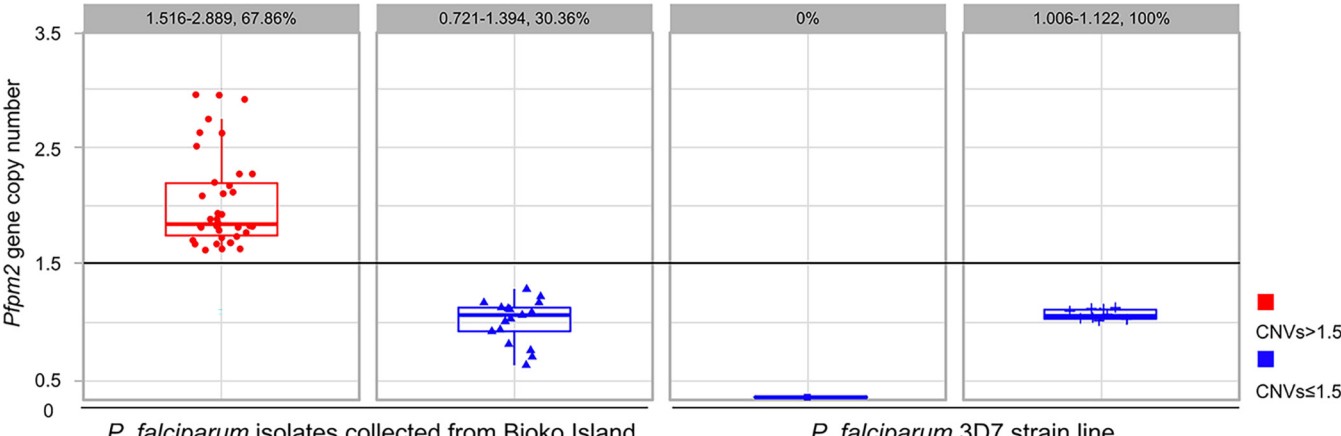

**FIG 1** *Pfpm2* gene copy number of *P. falciparum* isolates collected from Bioko Island and the *P. falciparum* 3D7 strain line. The results for *Pfpm2* gene amplification with Ct values ≥33 or CNVs <0.5 were considered invalid.

nonsynonymous substitutions per site (dN) explored the role of natural selection on *Pfk13* polymorphism (22). Furthermore, Tajima's D test was performed. The values of dS/dN and Tajima's D were −0.0004438 ($P > 0.05$) and −1.644 ($P > 0.05$), respectively. The values of both neutral tests were insignificant, indicating that the Bioko parasites have no bottleneck effect or rapid expansion and other historical events at the overall level.

**Genealogical relationships between Bioko and global *Pfk13*.** Comparative analysis of the global *Pfk13* gene illustrated that it is relatively conserved in the global parasite. We identified 5 to 595 *Pfk13* gene sequences with mutations at 1 to 39 loci in 14 countries (Fig. 2A). The mutation ratio (number of mutant/total number of sequences)

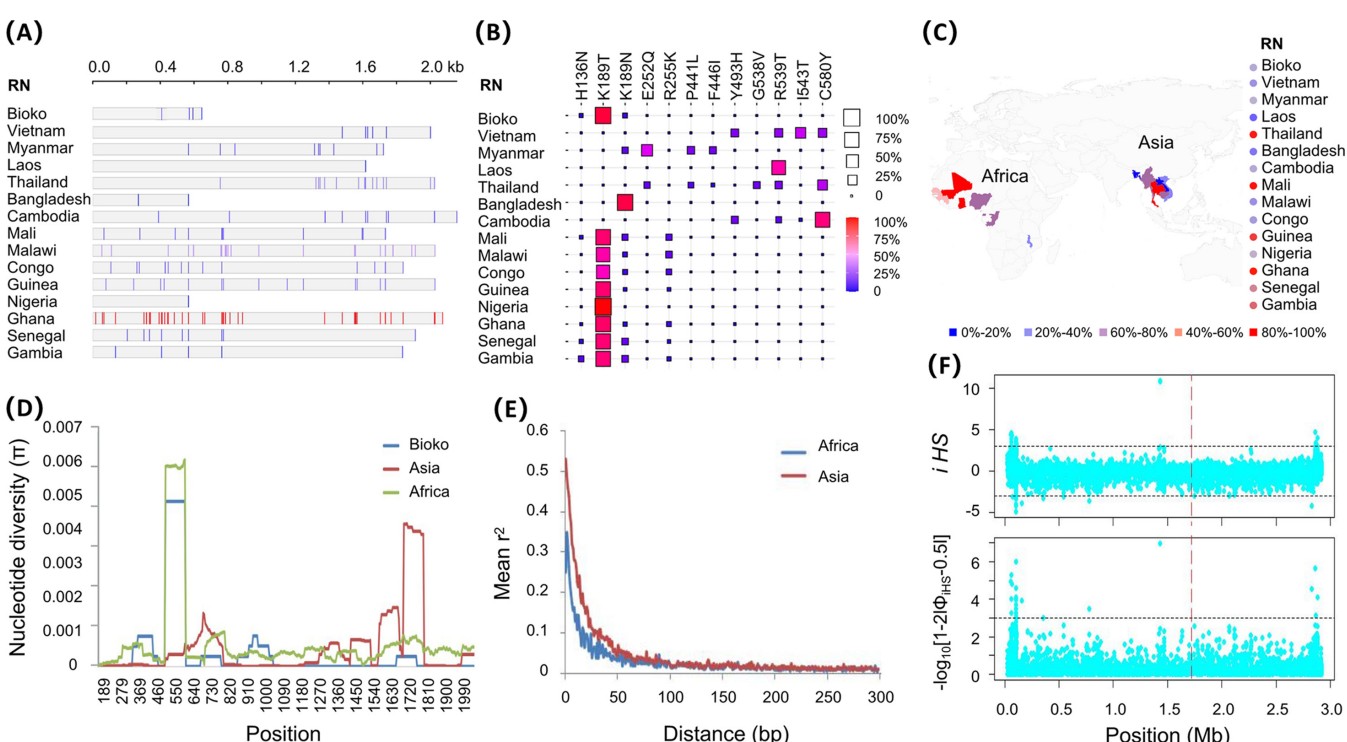

**FIG 2** Genealogical relationship analysis of Bioko and global *Pfk13*. (A) Number and localization of *Pfk13* mutation. (B) Mutation frequency of the *Pfk13* gene in different regions. The value is indicated by the square size and color: the larger the square and the redder the color, the higher the mutation frequency. (C) Mutation rates of the *Pfk13* gene in different regions. RN: region name. (D) Decay of linkage disequilibrium (LD) between Africa and Asia. (E) Sliding-window analysis of the nucleotide diversity in the global *Pfk13* gene. A window size of 100 bp and a step size of 5 bp were used. (F) A plot of integrated haplotype scores (iHS).

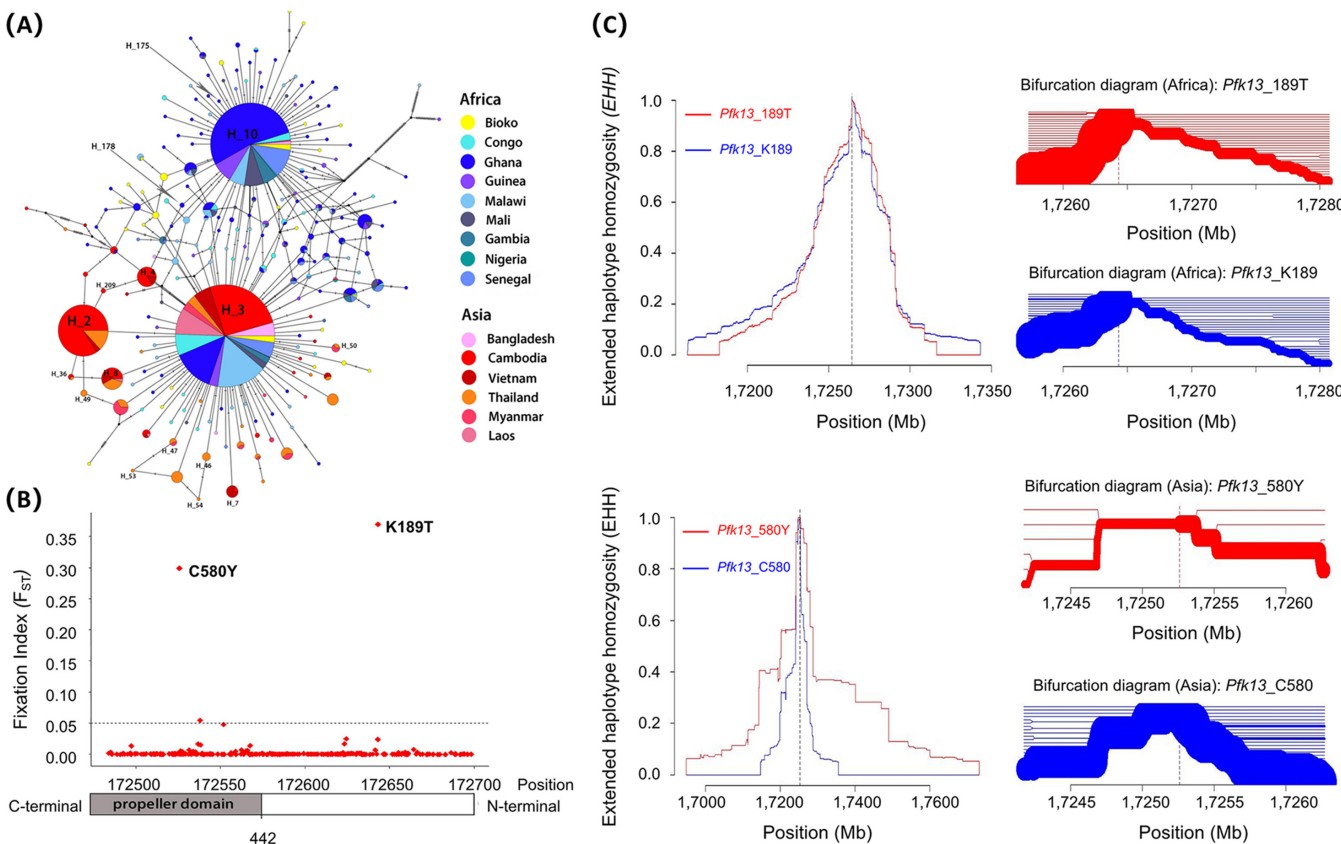

**FIG 3** (A) TCS network of the *Pfk13* haplotypes using 1418 infected cases from 14 countries and Bioko Island. H_3, haplotype contain the *Pfk13*_K189 allele; H_10, haplotypes contain *Pfk13*_189T allele; haplotypes contain *Pfk13*_446I allele (H_50); haplotypes contain *Pfk13*_458Y allele (H_46, H_54 and H_175); haplotypes contain *Pfk13*_493H allele (H_4, H_178 and H_209); haplotypes contain *Pfk13*_539T allele (H_8 and H_36); haplotype contain *Pfk13*_543T allele (H_7); haplotype contain *Pfk13*_561H allele (H_47 and H_53); haplotypes contain *Pfk13*_580Y allele (H_2, H_36 and H_49). (B) $F_{ST}$ between the African population and the Asian population. $F_{ST}$ scores were calculated for SNPs across the *Pfk13* gene; two SNPs with $F_{ST} > 0.2$ were labeled *Pfk13*_C580Y and *Pfk13*_K189T. The average $F_{ST}$ value for all SNPs of the *Pfk13* gene was 0.0054. (C) The decay of EHH in the region around *Pfk13*_K189T or *Pfk13*_C580Y: evidence from African or Asian populations.

exhibited a more pronounced African preference than Asian (Fig. 2C). These variants exhibit an uneven geographical distribution characterized by both loci and frequencies, for example, *Pfk13*_V6I and *Pfk13*_M18I were detected only from Malawi and Ghana. High-frequency mutations (>10%) at 12 loci were the predominant variation (11.29% to 88.89%), with a significantly higher frequency of mutations at locus 189 (Fig. 2B). Bioko and African isolates displayed high diversity in the nonpropeller domain of *Pfk13*, with largely overlapping $\pi$ curves. In contrast, the Asian isolates demonstrated extremely high diversity in the propeller domain (Fig. 2D). Tajima's D test was also performed to evaluate the natural selection of *Pfk13* in African and Asian parasites. The value was significant in Asian parasites ($-2.343$, $P < 0.05$) but not in African populations ($-2.710$, $P > 0.05$). Linkage disequilibrium (LD) decay results found that African parasites decayed more rapidly than Asian parasites, which was consistent with our result that African parasites exhibited higher diversity than Asian parasites. All these results suggested that Asian parasites have purifying selection, or as a signature of a recent population expansion upon the *Pfk13* gene (Fig. 2E). Long-range haplotype analysis was used to assess the effect of selection on the regions around the *Pfk13* gene in Asian and African populations. There is no evidence that any significant selective signature around the *Pfk13* gene was calculated by the |iHS| (Fig. 2F).

**Haplotype network analysis of global *Pfk13*.** The Templeton, Crandall, and Sing (TCS) network of the *Pfk13* gene is distributed using the two most dominant haplotypes, H_3, and H_10 radiation, which are shared by the parasites in Bioko, Africa, and Asia. The separation of H_3 and H_10 was caused by *Pfk13*_K189T (Fig. 3A). This

information can also be derived from the fixation index ($F_{ST}$) (Fig. 3B). The H_10 haplotype was distributed mainly in the African population and the others (H_3) in Asian populations. Although several unique haplotypes were detected from the Bioko parasites, the Bioko haplotypes were more closely related to the haplotypes in Africa than in Asia. Notably, the haplotypes containing artemisinin-resistant mutations were closely linked to H_3 haplotypes (H_3) carrying the wild-type allele (*Pfk13*_K189). *Pfk13*_C580Y was the most frequent artemisinin-resistant mutation in Asian parasites.

**Test for recent selection on *Pfk13*_K189T and *Pfk13*_C580Y.** Because the present data consisted of single nucleotide polymorphisms (SNPs) located only on the *Pfk13* gene, which lacks enough data on both sides of the core SNPs, a further extended haplotype homozygosity (EHH) analysis containing more information on the MalariaGEN Pf3k Project was developed to detect the molecular signature of recent selection on *Pfk13*_K189T and *Pfk13*_C580Y. To our knowledge, SNPs under neutral evolution take a long time to reach high frequencies, during which the LD around the allele decays substantially due to recombination. In contrast, the SNPs under positive selection will rise to a high frequency so rapidly that long-range associations with neighboring SNPs are not disrupted by recombination. In the present study, we observed that the EHH of SNPs decays was similar for haplotypes carrying *Pfk13*_K189 and *Pfk13*_189T ($P > 0.05$). However, *Pfk13*_580Y required more than 60.0 kb for EHH to decay to a level of 0.05, as opposed to 20.0 kb for *Pfk13*_C580. There were significant differences between them ($P < 0.05$). We plotted the haplotype bifurcation diagrams for the two mutations from these data (Fig. 3C). Only *Pfk13*_580Y was found to have an extended predominance by showing a thick branch in the haplotype-bifurcation diagram, clearly suggesting long-range linkage disequilibrium (LD). All the above results revealed that *Pfk13*_580Y was under strong positive selection in Asian parasites, but *Pfk13*_189T had not yet conferred a selective advantage over *Pfk13*_K189 in the survival of African parasites.

## DISCUSSION

There is a perennially high rate of malaria transmission throughout Bioko Island, and ACTs are currently commonly used for treatment. The Malaria National Treatment Guide of Equatorial Guinea recommended AS/AQ and AL as the first and second-line treatments for uncomplicated malaria, respectively. DHA/PPQ was also used routinely by some physicians from the public and private sectors in the region (23). Regular drug efficacy monitoring using therapeutic efficacy studies (TES) or drug-resistance molecular markers has been recommended by the WHO to allow early detection of emerging resistance and facilitate prompt policy changes before therapeutic efficacy falls below 90%. Routine monitoring of the efficacy of recommended ACTs at least every 2 years is also proposed. From 2017 to 2018, AS/AQ and AL were shown to be highly effective in the treatment of uncomplicated *P. falciparum* infection in Equatorial Guinea (Bioko, Annobo'n, and Corisco Bay) with cure rates of 98.6% to 100% and 92.4% to 100%, respectively. A small proportional mutation of *Pfk13* occurred which was not associated with artemisinin resistance (24). The data presented in this study are the first molecular investigations conducted for the *Pfcrt*, *Pfk13*, *Pfpm2*, and *Pfmdr1* genes on Bioko Island and assessed the polymorphic profile of the *Pfk13* gene. Accordingly, we aimed to offer valuable references for the assessment of resistance to ACT partner drugs by monitoring molecular markers associated with the drug resistance.

Sample collection for this study was carried out in Malabo Regional Hospital, Bioko Island. Bioko, with an area of approximately 2,000 square kilometers, is an island 32 km off the west coast of Africa and its population is approximately 330,000 (2015 census) (25). Because of its geographic posture of active volcanoes and tropical rainforest, approximately 90% of the population lives in the capital, Malabo, with only a small percentage living in small villages outside the city. Serious illnesses (including malaria) in these areas re generally cared for at the Malabo Regional Hospital, a government-owned public hospital (26). Therefore, the sample we collected is somewhat geographically

representative. A previous study analyzed the genetic diversity of merozoite surface proteins 1 (MSP-1) and 2 (MSP-2) in *P. falciparum* from Malabo Regional Hospital. The results showed that the MAD20 (9 alleles) family dominated in *msp1*, followed by the K1 (9 alleles) and R033 (8 alleles) families. In *msp2*, the FC27 (5 alleles) family was the most frequently detected, followed by the 3D7 (20 alleles) family (27). Although the geographical area of Bioko Island is small, mixed clones of *P. falciparum* are still present, which further indicates that the *Plasmodium* parasites in the region are geographically representative. Unfortunately, the limited period and the number of samples collected, coupled with the increased difficulty of sample collection due to the novel coronavirus epidemic. Therefore, further exploration between time and resistance is hampered, which will be the focus of our later studies.

*P. falciparum's* resistance to artemisinin and its derivatives has spread in Southeast Asia and is threatening malaria control efforts in sub-Saharan Africa (28). Mutations in the propeller domain of *Pfk13* have become a molecular marker for detecting artemisinin resistance. The mutation (*Pfk13*_M579I) has been found in a Chinese migrant worker returning from Equatorial Guinea, but studies have not yet detected *Pfk13* mutations linked with artemisinin resistance in the local population (29). In addition, another *Pfk13* mutation (*Pfk13*_R561H) associated with artemisinin resistance has been identified in Africa (30). Bioko *Pfk13* resistance mutations were not observed in this study, and no significant selection effect was obtained by using bioinformatics tools. Previous studies have shown that only three Pfk3 mutations (*Pfk13*_E433D and *Pfk13*_A578S) were present in 476 samples after AS/AQ and AL treatment in the Equatorial Guinea region (including Bioko) and were also not associated with known artemisinin resistance-related markers (24). These results demonstrated that artemisinin remains highly effective and supported the BIMEP recommendation above.

Similar to other African regions, such as Senegal (31), the *Pfk13*_K189T mutation was observed at a high frequency in Bioko parasites. Consistent with the geographical distribution of Bioko, the TCS network diagram showed that most of the haplotypes from Bioko were more closely related to the African population than the Asian population. More strikingly, most of the haplotypes that carried validated *Pfk13* resistance mutations were shown to originate from the haplotypes carrying the *Pfk13*_K189 allele (wild-type). Although *Pfk13*_K189T did not exhibit signals of selection in Africa and was not correlated with delayed parasite clearance, continued research on *Pfk13*_K189T in Africa might be important for understanding the phenomenon that artemisinin resistance developed more slowly in Africa than in other regions.

In Africa, treatment failure with ACTs has occurred mainly because of partner drug resistance. Mutations of *Pfmdr1* and *Pfcrt* have been reported to be associated with *in vitro* responses to two partner drugs: amodiaquine and lumefantrine (32). Previous studies showed that the *Pfmdr1*_N86 and *Pfcrt*_K76 mutant alleles were associated with lumefantrine resistance and DHA resistance, whereas the *Pfmdr1*_86Y and *Pfcrt*_76T wild-type alleles were significantly associated with decreased susceptibility to amodiaquine. A meta-analysis of African individual patient data revealed that *Pfmdr1*_N86 and *Pfcrt*_K76 parasites predominated (>80%), and AS/AQ provided ~2-fold longer protection than AL. Conversely, AL protected up to 1.5-fold longer than AS/AQ in the presence of high parasite prevalence (>80%) in the *Pfmdr1*_86Y and *Pfmdr1*_76T mutants (33). In this study, we observed prevalence rates of *Pfmdr1*_86Y, *Pfmdr1*_184F, and *Pfcrt*_76T in Bioko parasites of 22.69%, 43.70%, and 26.95%, respectively. They showed a decreasing trend in frequency compared with previous findings (*Pfmdr1*_86Y: 50.32%, *Pfmdr1*_184F: 89.81%, and *Pfcrt*_76T: 92.05%) (2011 to 2014) (34). A decrease in amodiaquine-resistance *Pfmdr1*_86Y and *Pfmdr1*_76T favors AS/AQ but increases lumefantrine-resistance *Pfmdr1*_N86 and the highly prevalent *Pfmdr1*_184F threatens AL and DHA/PPQ.

Multiple copies of the *Pfpm2* gene are associated with reduced sensitivity of *P. falciparum* parasites to PPQ *in vitro* and with DHA/PPQ failures in Cambodia (35). Data from Vietnamese clinical sites (Gai Lai and Binh Phuoc) indicated that amplification of

the *Pfpm2* gene is spreading in regions located along with Cambodia (36). Surprisingly, among the African isolates, the situation was sharply contrasted with Southern Asia. Data reported by Leroy et al. (37) revealed that high proportions of isolates with multiple copies of the *Pfpm2* gene were frequently observed in the African sites, especially in Burkina Faso and Uganda (>30%). A total of 30.36% of isolates had multiple copies of the *Pfpm2* gene observed on Bioko Island. In terms of chemoprevention, DHA/PPQ benefits from the long biological half-life of PPQ, which indicates that the isolates without *Pfk13* resistance mutations, but carrying multiple copies of *Pfpm2* might reduce drug efficacy. To date, there is no conclusive evidence that amplification of the *Pfpm2* gene is necessary for the development of resistance to PPQ in Africa (38). Therefore, it is still unclear whether clinical resistance to PPQ has existed on Bioko Island. This indicated that *in vitro* or *ex vivo* drug susceptibility assays will be required to assess the efficacy of DHA/PPQ treatment for uncomplicated malaria on Bioko Island in the future.

In conclusion, no artemisinin resistance genetic background was observed in our study, which suggests that there is no immediate threat to artemisinin efficacy on Bioko Island. However, the prevalence of *Pfmdr1*_N86Y and *Pfmdr1*_N184F mutations and multiple copies of the *Pfpm2* gene were still present. Based on our data, it is recommended that continued monitoring of molecular markers of resistance is necessary for Bioko Island, which might provide baseline prevalence data to guide the use of ACTs. Furthermore, in conjunction with the results of molecular monitoring, additional *ex vivo* susceptibility and ongoing TES are necessary to further confirm the greater suitability of artemisinin partner drugs such as amodiaquine, piperaquine, and lumefantrine.

## MATERIALS AND METHODS

**Study area.** The study was carried out in the Malabo Regional Hospital and the clinic of the Chinese medical aid team in the Republic of Equatorial Guinea on Bioko Island, Equatorial Guinea. Bioko is an island 32 km off the west coast of Africa and the northernmost part of Equatorial Guinea. Its population of approximately 334,463 (2015 census, of which approximately 90% live in Malabo, the capital city) is at risk of malaria year-round. Since the launch of the BIMCP in 2004, the parasite prevalence on Bioko Island decreased from more than 45% in 2004 to 8.5% in 2016, and the entomological inoculation rate decreased from more than 1,000 before 2004 to 14 in 2015 (27).

**Ethics approval and informed consent.** Ethical approval was obtained from the Ethics Committee of Malabo Regional Hospital in Malabo, Equatorial Guinea, and the Ethics Committee of Chaozhou People's Hospital Affiliated with Shantou University Medical College. All studies were conducted according to the ethical guidelines set by the aforementioned review boards. All adults who participated in these studies gave informed consent. All children involved in the studies gave consent and had parental consent to participate.

***Plasmodium falciparum* isolates.** A total of 152 patients with uncomplicated malaria were enrolled and analyzed between January 2017 and December 2019. The included patients were between 12 and 67 years of age and were residents of Bioko Island. Malaria patients were classified into uncomplicated malaria states according to the WHO criteria, which were defined as a positive smear for *P. falciparum* and the presence of fever (≥37.5°C) (39). Consent was obtained from all participating subjects or their parents. Laboratory screening for malaria was performed using an immunochromatographic diagnostic test (ICT Malaria Combo Cassette Test) and confirmed using microscopic examination of blood smears (10, 28). Extra blood drops were collected for a malaria smear and a Whatman 903 filter paper sample (GE Healthcare, Pittsburgh, USA). *Plasmodium* species were identified by real-time PCR followed by high-resolution melting (PCR-HRM) as in our previous reports (40).

**DNA extraction.** Genomic DNA was extracted from dried filter blood spots (DBS) with the Genomic DNA Extraction Kit for Dry Blood Spot ([no. DP334] TIANGENE Biotech [Beijing] CO., Ltd.), following the manufacturer's protocol. The extracted DNA was quantified with a NanoDrop2000 spectrophotometer (Thermo Fisher Scientific, USA), and then stored at −20°C until tested by PCR-Sequencing and PCR-HRM analysis.

**PCR amplification and sequencing of *Pfmdr1*, *Pfcrt*, and *Pfk13*.** We synthesized classical primers targeting *P. falciparum* for nested PCR as previously described (Table 2) (31, 32, 41). TaKaRa Taq™ HS Perfect Mix (TaKaRa, Carlsbad, CA) was used as the master mix and was supplemented with a 0.2 $\mu$M concentration of each primer. The final volumes for primary and nested PCR were 25 $\mu$L (12.5 $\mu$L master mix plus 1 $\mu$L DNA template) and 50 $\mu$L (25 master mix plus l $\mu$L primary PCR product), respectively. Reaction system amplification of the primary round was carried out under the following conditions: 95°C for 3 min; 30 cycles of 98°C for 10 sec, 58°C for 10 sec, and 72°C for 2 min; and a final extension at 72°C for 10 min. The PCR products obtained from the primary round were used as DNA templates for the second round of amplification. The reaction conditions for the second round were as follows: 95°C for 3 min; 35 cycles of 98°C for 10 sec, 55°C for 5 sec, and 72°C for 2 min; and a final extension at 72°C for 10 min.

All PCR products were analyzed by 1.0% agar gel electrophoresis, and DNA sequencing was performed using an ABI 3730XL automated sequencer (PE Biosystems, CT, USA). The nucleotide and

**TABLE 2** Primers for genotyping *Pfmdr1*, *Pfcrt*, *Pfk13*, and *Pfpm2*

| Gene (ID) | PCR round | Primer name | Sequence (5'–3') | Size (bp) |
|---|---|---|---|---|
| *Pfmdr1* | 1st round PCR | *Pfmdr1*-F1 | TTAAATGTTTACCTGCACAACATAGAAAATT | 612 |
| (PF3D7_0523000) | | *Pfmdr1*-R1 | CTCCACAATAACTTGCAACAGTTCTTA | |
| | 2nd round PCR | *Pfmdr1*-F2 | TGTATGTGCTGTATTATCAGGA | |
| | | *Pfmdr1*-R2 | CTCTTCTATAATGGACATGGTA | 526 |
| *Pfcrt* | 1st round PCR | *Pfcrt*_F1 | CCGTTAATAATAAATACACGCAG | 547 |
| (PF3D7_0709000) | | *Pfcrt*_R1 | CGGATGTTACAAAACTATAGTTACC | |
| | 2nd round PCR | *Pfcrt*_F2 | TGTGCTCATGTGTTTAAACTT | 145 |
| | | *Pfcrt*_R2 | CAAAACTATAGTTACCAATTTTG | |
| *Pfk13* | 1st round PCR | *Pfk13*_F1 | GGGAATCTGGTGGTAACAGC | 2097 |
| (PF3D7_1343700) | | *Pfk13*_R1 | CGGAGTGACCAAATCTGGGA | |
| | 2nd round PCR | *Pfk13*_F2 | GCCAAGCTGCCATTCATTTG | 2027 |
| | | *Pfk13*_R2 | GCGGAAGTAGTAGCGAGAAT | |
| *Pfpm2* | | *Pfpm2*_F | TGGTGATGCAGAAGTTGGAG | 79 |
| (PF3D7_1408000) | | | | |
| | | *Pfpm2*_R | TGGGACCCATAAATTAGCAGA | |
| *β-tubulin* | | *β-tubulin*_F | TGATGTGCGCAAGTGATCC | 79 |
| | | *β-tubulin*_R | TCCTTTGTGGACATTCTTCCTC | |

deduced amino acid sequences of *Pfk13* were analyzed using EditSeq and SeqMan in the DNASTAR package (DNASTAR, Madison, WI, USA). The *Pfk13*, *Pfmdr1*, and *Pfcrt* sequences of the laboratory-adapted *P. falciparum* strain 3D7 (XM_001351086) were included in the alignment for comparison as a reference sequence.

**Pfpm2 gene copy number variation assessment.** The copy number of the *Pfpm2* (PF3D7_1408000) gene was measured using a SLAN-96S real-time PCR instrument (Shanghai, China), relative to a single copy of the *β-tubulin* gene (used as a reference gene). The primers and experimental procedures used for qPCR were as described by Didier Ménard et al. (42), with minor changes (Table 2). The reaction mixture for qPCR was as follows: 10 $\mu$L 2× *Taq* PCR MasterMix, 2 $\mu$L 20× EvaGreen Dye (Biotium), 0.6 $\mu$L of each primer, 4 $\mu$L of template DNA, and $H_2O$ to a final volume of 20 $\mu$L. The *P. falciparum* 3D7 strain line was used as a control and was presented by Jun Cao from the Jiangsu Institute of Parasitic Diseases, China. Six replicates were included in each run, with amplification of each sample performed in three replicates. After amplification, the specificity of *Pfpm2* was verified by melting curves and *Pfpm2* copy numbers were calculated by the $2^{-\Delta\Delta Ct}$ method. A *Pfpm2* copy number >1.5 was defined as an amplification of the gene (37).

**Genetic diversity analysis and natural selection.** To estimate the genetic diversity of the *Pfk13* gene, polymorphism analysis of DNA sequences was performed on 96 *Pfk13* sequences from Bioko Island. Sequence alignment was performed by using the CLUSTAL W program in MEGA7.0 (43). The number of haplotypes (H), haplotype diversity (Hd), nucleotide diversity ($\pi$), and an average number of pairwise nucleotide differences (K) of the *Pfk13* gene were calculated using DnaSP (version 6.1) (44). To explore the natural selection of *Pfk13*, values of nonsynonymous (dN) and synonymous (dS) substitutions were estimated and compared using the Z test ($P < 0.05$ was significant) in the MEGA7.0 program based on the Nei-Gojobori method with Jukes-Cantor correction (45). Tajima's D values were analyzed using DnaSP version 6.1 to evaluate the neutral theory of evolution.

**Haplotype network and genetic differentiation.** To assess the genealogical relationships between global *Pfk13* haplotypes and Bioko haplotypes, we retrieved all genetic variants on chromosome 13 available from the MalariaGEN Pf3k Project (release 5, https://www.malariagen.net/parasite/pf3k) in variant call format (VCF). Variants were called using SAMtools (46) and VCFtools (47) with default settings. Variants in the entire *Pfk13* gene were extracted for the 1,322 infected cases from Africa and Asia (48). The Templeton, Crandall, and Sing (TCS) method in PopArt32 (http://popart.otago.ac.nz/index.shtml) was used to establish genealogical relationships among global *Pfk13* haplotypes. VCFtools was used to compare the sequences between African and Asian countries in terms of diversity by calculating $F_{ST}$ (49).

**Extended haplotype homozygosity analysis.** Extended haplotype homozygosity analysis associated with *Pfk13*_K189T and *Pfk13*_C580Y was performed on global *P. falciparum* isolates using the R package REHH 2.0 (50). The EHH of *Pfk13*_K189T in the African population and *Pfk13*_C580Y in the Asian population were calculated using SNPs with a minor allele frequency >0.05.

**Statistical analysis.** All statistical analyses were performed using SPSS version 19.0 software (SPSS Inc., Chicago, IL, USA). A *P* value < 0.05 was considered statistically significant.

## ACKNOWLEDGMENTS

We thank the Department of Health of Guangdong Province and the Department of Aid to Foreign Countries of the Ministry of Commerce of the People's Republic of China for their help. We also thank Santiago-m Monte-Nguba for his technical help during the sample collection and diagnosis.

This research was funded by the Guangxi Provincial Natural Science Foundation (no. 2019JJD140052 and 2020JJA140656), Education Department Project of Guangdong Province (no. 2019-GDXK-0031 and 2020KZDZX1146), Eastern Guangdong Technological Engineering Research Center (no. P19004), Doctor Initiating Project of the Hanshan Normal University (no. QD202125 and QD20190527), Guangdong Provincial Key Laboratory of Functional Substances in Medicinal Edible Resources and Healthcare Products (no. 2021B1212040015) and the Principal Investigator Program of Hubei University of Medicine (no. HBMUPI202101).

We declare no conflict of interest.

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
