## [Reviewer comments · Microbiology Spectrum]

Microbiology Spectrum

Molecular Surveillance of Artemisinin-based Combination Therapies Resistance in *Plasmodium falciparum* Parasites from Bioko Island, Equatorial Guinea

YaQun Liu, Xue Liang, Jian Li, Jiangtao Chen, HuiYing Huang, YuZhong Zheng, JinQuan He, Carlos Ehapo, Urbano Eyi, PeiKui Yang, Li Lin, Wei Chen, GuangYu Sun, Xiang Liu, GuangCai Zha, JunLi Wang, ChunFang Wang, HuaGui Wei, and Min Lin

Corresponding Author(s): Min Lin, Hanshan Normal University

Review Timeline:

Submission Date:	February 2, 2022
Editorial Decision:	March 11, 2022
Revision Received:	March 28, 2022
Editorial Decision:	April 20, 2022
Revision Received:	April 25, 2022
Accepted:	May 10, 2022

Editor: Laura Kirkman

Reviewer(s): Disclosure of reviewer identity is with reference to reviewer comments included in decision letter(s). The following individuals involved in review of your submission have agreed to reveal their identity: Mateuz Placinski (Reviewer #1); Oriana Kreutzfeld (Reviewer #2)

Transaction Report:

DOI: <https://doi.org/10.1128/spectrum.00413-22>

March 11, 2022

Dr. Min Lin
Hanshan Normal University
School of Food Engineering and Biotechnology
chaozhou
China

Re: Spectrum00413-22 (Molecular Surveillance of Artemisinin-based Combination Therapies Resistance in *Plasmodium falciparum* Parasites from Bioko Island, Equatorial Guinea)

Dear Dr. Min Lin:

Link Not Available

Sincerely,

Laura Kirkman

Journals Department
Reviewer comments:

Reviewer #1 (Comments for the Author):

Paper Summary

Lui et al describe the molecular markers of ACT resistance in a small convenience sample of approximately 100 patients with uncomplicated malaria at Malabo Regional Hospital from Jan 2017 - Dec 2019.

Major comments

Epi Methods

1. Samples collected over 2 years (2017-2019), this should be discussed as a limitation, as resistance could have changed over this time. Would recommend a sensitivity analysis to explore the effect of time. Break it up in years or 6-month buckets or another method. Is there a difference over time?
2. Lack of geographical representation. This should be discussed as a limitation. It appears that all samples came from Malabo Regional Hospital and Clinic of the Chinese Medical Aid Team. It is not clear if the results of this convenience sample can be generalized to the overall parasite population on the island
3. Results should be compared against a recent Therapeutic Efficacy Study (TES) that was not referenced. One of the three sites was Malabo. (PMID: 34158055). Do these molecular markers of resistance correlate with clinical efficacy from this recent TES?

Molecular Methods

1. Why were only monoclonal samples evaluated? What was done with the polyclonal sequences?
2. How were monoclonal infections identified?
3. Figure 1 - would be clearer presented as a table. Recommend including the gels (PFE) in a supplement.
4. Figure 2 - Standard box plots would more clearly represent the data. The violin plots incorrectly show a tail of the distribution below the threshold in the left-most plot.
5. Pfk13 Haplotype analysis: This analysis is very similar to a previously published report from 2019 on pfk13 diversity. PMID: 31085516. It appears to differ only by inclusion of the Bioko samples. Worryingly, almost entire sentence is copied and pasted verbatim from this manuscript (for example, lines 140-141 and lines 171-173). Notably, the earlier paper (PMID: 31085516) is not cited in this paper. Beyond the apparent plagiarism, it is not clear how much inclusion of Bioko samples adds to this previous analysis.

Conclusions

1. Appropriateness of conclusions is overstated. It is not appropriate to recommend changing ACT therapy solely based molecular data from only approximately 100 samples without data on clinical efficacy, such as from a TES. The recent TES in Equatorial Guinea was not referenced and the results of this study were not interpreted in the context of this important source of efficacy data. Does the genotypic data from this study match the phenotypic data (recent TES)?

Minor comments

1. Line 32: Prevalence of what? Mutations? Repeats?
2. Line 38: "high efficacy" can only be determined with a TES.
3. Line 110: Describe why only 112 sequences were successfully obtained? Were they polyclonal?
4. Line 113: Cite references that show those mutations were not associated with artemisinin delayed clearance.
5. Line 136: Why were only 96/152 samples evaluated for Pfk13?
6. Fig 3: Adding Bioko data to A, B, C would be helpful.
7. Line 171 and 182: Spell out acronyms at first use.
8. Line 218: Prior TES have been completed in Equatorial Guinea (including Malabo) and should be described and referenced. Results from TES and this study should be compared and contrasted.
9. Line 256: Leroy D? Is this a reference? If so, needs appropriate reference.

Reviewer #2 (Comments for the Author):

The work by Liu et al. investigates the presence of mutations in PfKelch13, the marker associated with resistance to current artemisinin based antimalarials, and known resistance markers for partner drugs like amodiaquine and piperazine in a region in Bioko Island in Equatorial Guinea from 2017 to 2019. The findings indicate that there is little polymorphism in Pfk13 on Bioko Island with one frequently observed mutation Pfk_K189T that is not associated with artemisinin resistance. Prevalence of Pfcrt and Pfmdr1 mutations associated with resistance to amodiaquine, piperazine and lumefantrine were fairly high indicating that artemisinin partner drug efficacy might be challenged on Bioko Island.

Current recommendations to treat malaria by the WHO are artemisinin combination therapies. Artemisinin resistance is prominent in Asia and first reports suggest association of specific PfKelch13 mutations with delayed parasite clearance in Africa. Surveillance of Pfk13 mutations in African region is of great important. Liu et al. report on surveillance of Pfk13 mutations and partner drugs resistance markers Pfcrt and Pfmdr1 on Bioko Island in Equatorial Guinea. The results suggest little polymorphisms in Pfk13, but presence of resistance markers in Pfmdr1 and Pfcrt. The study adds to the understanding of circling Pfk13 resistance markers in Africa and comparisons with other isolates from Africa and Asia highlight the separate emergence of Pfk13 mutations in Africa and Asia.

Comments:

1. The manuscript is well written and conveys the messages clearly and concisely, however, there are several grammatical mistakes, words missing and abbreviations not explained. I suggest looking closely at the manuscript and fixing these issues. A few examples: Line 89, ACTs mentioned twice, line 94 should be Bioko Island reported

2. The abstract and importance section need to be overhauled in terms of grammatical, spelling errors and the usage of the wrong verbs and adjectives. Additionally, the grammatical errors make it difficult to understand which message the authors want to convey and can therefore, not be properly evaluated.
3. Line 32: The authors state that prevalence of Pfk13, Pfcr1 and Pfmdr1 was 73.68%, 78.29% and 75.66%. Since these are genes that are always present in *P. falciparum* (prevalence 100%) what are the numbers referring to? Are these numbers based on a specific mutation?
4. Line 96: The manuscript states that the efficacy of DHA/PPQ has not been documented on Bioko Island. Is this also true for the other ACTs?
5. Line 101: The last sentence is a conclusive statement that is not backed up by the data analyzed in the manuscript. To decide on treatment options for malaria patients, additional clinical efficacy studies of the drugs would need to be conducted. Consider revising this sentence to better reflect the outcome of this study.
6. Line 104: The first paragraph could be combined with the following paragraph.
7. Line 112: The sentence states that none of the observed mutations was associated with delayed parasite clearance. Does this refer to previous studies or was this evaluated in this study? For the former, please revise sentence to reflect the reference to former studies and reference them, if the latter, how was this evaluated? Where IC50 data obtained for the isolates?
8. Line 116-117: The sentence talks about the Pfmdr1 and Pfcr1 mutations, but the reference to the figure is missing. Please indicate the correct figure reference here.
9. Figure 1: The figure shows the mutations found in pfmdr1, pfcr1 and pfk13. The figure does not include the sequence change for pfcr1_K76T. Please describe the schematic gene map and the colors/symbols in the figure legend. C is not mentioned in the text and is not informative for the reader and should be removed from the main figure. Additionally, electropherograms for Pfk13_N194D and Pfk13_S213G are of poor quality and actually showing mixed infection for these two mutations. This should be reflected in the text and analysis.
10. Line 126: The authors state that they used a threshold of 1.5 to define single or multiple copies of Pfpm2. How was this threshold determines? It would be more accurate to have a "positive" control with 2 copies of Pfpm2. I would suggest running a few additional qPCRs with a 2 copy and a 1 copy control to precisely determine the threshold.
11. Line 129-130: The authors speculate that due to increase in Pfpm2 copy number variation drug resistance has emerged in Bioko Island. This is a bold statement that is only made on Pfpm2 copy number variation, experimental susceptibility assays or clinical data is missing. Therefore, I suggest rephrasing this sentence and using more cautious words to describe the potential for drug resistance. Additionally, it would be good to state which drug would be effected.
12. The manuscript uses many different biostatistical analyses for the relatedness and genetic diversity of the isolates. It is important to clarify all the abbreviations used to describe these analyses. Examples are: Line 136: "K value", line 139 "Hd" + " π ", Line 141 "dS" and "dN", line 157 "LD", Line 174 "FST index", Line 198 "TCS", Line 204 "FST", Line 182 "EHH"
13. Figure 3: Figure 3 shows the mutation frequency in Pfk13 in African and Asian parasites. Figure 3B: It would be good to describe the frequency scale in the figure legend as there are two different parameters (color and size of square) listed for the mutation frequency.
14. Line 210: The authors state that there is a high rate of malaria transmission on Bioko Island, however, in the introduction it is stated that the transmission rate went down from 45% to 8.5%. Can you elaborate on this?
15. Line 218: The authors state that the study is looking at the efficacy of ACTs by looking at molecular markers. To determine efficacy of the drugs the study would have needed to include either susceptibility data of isolates or clinical efficacy data. I advice to revise this sentence to adequately reflect the topic of the paper e.g. "The data presented here describe the first study on Bioko Island looking at molecular markers associated with ACT drug resistance".
16. Line 233-236: The conclusion for the Pfk13_K189T found in Bioko parasites is that it does not influence artemisinin resistance; but that it might be good to study this mutation further to understand why artemisinin resistance is developing more slowly in Africa. How does this mutation have an influence on artemisinin resistance development? Can you elaborate further on this speculation?
17. Line 249: The authors state that the pfmdr1 ad pfcr1 mutation frequency decreased compared to previous findings. However, the authors also noted that this is the first study on Bioko Island. Can you clarify this point?
18. References are overall appropriate for the topic, however, line 253-258 are missing two references.
19. The final conclusion of the paper sates that „ Based on our data it is speculated that amodiaquine may be the better artemisinin partner drug than piperazine and lumefantrine on Bioko Island." I suggest that another sentence will be added calling for additional ex vivo susceptibility and clinical efficacy studies to adequately study this speculation.
20. The authors performed sequencing for Pfk13, Pfcr1 and Pfmdr1. The sequencing data should be made available on the NCBI BankIT database.

Staff Comments:

Preparing Revision Guidelines

To submit your modified manuscript, log onto the eJP submission site at <https://spectrum.msubmit.net/cgi-bin/main.plex>. Go to Author Tasks and click the appropriate manuscript title to begin the revision process. The information that you entered when you first submitted the paper will be displayed. Please update the information as necessary. Here are a few examples of required

updates that authors must address:

Please return the manuscript within 60 days; if you cannot complete the modification within this time period, please contact me. If you do not wish to modify the manuscript and prefer to submit it to another journal, please notify me of your decision immediately so that the manuscript may be formally withdrawn from consideration by Microbiology Spectrum.

The work by Liu et al. investigates the presence of mutations in PfKelch13, the marker associated with resistance to current artemisinin based antimalarials, and known resistance markers for partner drugs like amodiaquine and piperazine in a region in Bioko Island in Equatorial Guinea from 2017 to 2019. The findings indicate that there is little polymorphism in Pfk13 on Bioko Island with one frequently observed mutation Pfk_K189T that is not associated with artemisinin resistance. Prevalence of Pfcrt and Pfmdr1 mutations associated with resistance to amodiaquine, piperazine and lumefantrine were fairly high indicating that artemisinin partner drug efficacy might be challenged on Bioko Island.

Current recommendations to treat malaria by the WHO are artemisinin combination therapies. Artemisinin resistance is prominent in Asia and first reports suggest association of specific PfKelch13 mutations with delayed parasite clearance in Africa. Surveillance of Pfk13 mutations in African region is of great important. Liu et al. report on surveillance of Pfk13 mutations and partner drugs resistance markers Pfcrt and Pfmdr1 on Bioko Island in Equatorial Guinea. The results suggest little polymorphisms in Pfk13, but presence of resistance markers in Pfmdr1 and Pfcrt. The study adds to the understanding of circling Pfk13 resistance markers in Africa and comparisons with other isolates from Africa and Asia highlight the separate emergence of Pfk13 mutations in Africa and Asia.

Comments:

1. The manuscript is well written and conveys the messages clearly and concisely, however, there are several grammatical mistakes, words missing and abbreviations not explained. I suggest looking closely at the manuscript and fixing these issues. A few examples: Line 89, ACTs mentioned twice, line 94 should be Bioko Island reported
2. The abstract and importance section need to be overhauled in terms of grammatical, spelling errors and the usage of the wrong verbs and adjectives. Additionally, the grammatical errors make it difficult to understand which message the authors want to convey and can therefore, not be properly evaluated.
3. Line 32: The authors state that prevalence of Pfk13, Pfcrt and Pfmdr1 was 73.68%, 78.29% and 75.66%. Since these are genes that are always present in *P. falciparum* (prevalence 100%) what are the numbers referring to? Are these numbers based on a specific mutation?
4. Line 96: The manuscript states that the efficacy of DHA/PPQ has not been documented on Bioko Island. Is this also true for the other ACTs?
5. Line 101: The last sentence is a conclusive statement that is not backed up by the data analyzed in the manuscript. To decide on treatment options for malaria patients, additional clinical efficacy studies of the drugs would need to be conducted. Consider revising this sentence to better reflect the outcome of this study.
6. Line 104: The first paragraph could be combined with the following paragraph.
7. Line 112: The sentence states that none of the observed mutations was associated with delayed parasite clearance. Does this refer to previous studies or was this evaluated in this study? For the former, please revise sentence to reflect the reference to former studies and reference them, if the latter, how was this evaluated? Where IC50 data obtained for the isolates?
8. Line 116-117: The sentence talks about the Pfmdr1 and Pfcrt mutations, but the reference to the figure is missing. Please indicate the correct figure reference here.
9. Figure 1: The figure shows the mutations found in pfmdr1, pfcrt and pfk13. The figure does not include the sequence change for pfcrt_K76T. Please describe the schematic gene map and the colors/symbols in the figure legend. C is not mentioned in the text

and is not informative for the reader and should be removed from the main figure. Additionally, electropherograms for Pfk13_N194D and Pfk13_S213G are of poor quality and actually showing mixed infection for these two mutations. This should be reflected in the text and analysis.

10. Line 126: The authors state that they used a threshold of 1.5 to define single or multiple copies of Pfp_{m2}. How was this threshold determined? It would be more accurate to have a “positive” control with 2 copies of Pfp_{m2}. I would suggest running a few additional qPCRs with a 2 copy and a 1 copy control to precisely determine the threshold.
11. Line 129-130: The authors speculate that due to increase in Pfp_{m2} copy number variation drug resistance has emerged in Bioko Island. This is a bold statement that is only made on Pfp_{m2} copy number variation, experimental susceptibility assays or clinical data is missing. Therefore, I suggest rephrasing this sentence and using more cautious words to describe the potential for drug resistance. Additionally, it would be good to state which drug would be affected.
12. The manuscript uses many different biostatistical analyses for the relatedness and genetic diversity of the isolates. It is important to clarify all the abbreviations used to describe these analyses. Examples are: Line 136: “K value”, line 139 “Hd” + “ π ”, Line 141 “dS” and “dN”, line 157 “LD”, Line 174 “F_{ST} index”, Line 198 “TCS”, Line 204 “F_{ST}”, Line 182 “EHH”
13. Figure 3: Figure 3 shows the mutation frequency in Pfk13 in African and Asian parasites. Figure 3B: It would be good to describe the frequency scale in the figure legend as there are two different parameters (color and size of square) listed for the mutation frequency.
14. Line 210: The authors state that there is a high rate of malaria transmission on Bioko Island, however, in the introduction it is stated that the transmission rate went down from 45% to 8.5%. Can you elaborate on this?
15. Line 218: The authors state that the study is looking at the efficacy of ACTs by looking at molecular markers. To determine efficacy of the drugs the study would have needed to include either susceptibility data of isolates or clinical efficacy data. I advice to revise this sentence to adequately reflect the topic of the paper e.g. “The data presented here describe the first study on Bioko Island looking at molecular markers associated with ACT drug resistance”.
16. Line 233-236: The conclusion for the Pfk13_K189T found in Bioko parasites is that it does not influence artemisinin resistance; but that it might be good to study this mutation further to understand why artemisinin resistance is developing more slowly in Africa. How does this mutation have an influence on artemisinin resistance development? Can you elaborate further on this speculation?
17. Line 249: The authors state that the pfmdr1 and pfcrt mutation frequency decreased compared to previous findings. However, the authors also noted that this is the first study on Bioko Island. Can you clarify this point?
18. References are overall appropriate for the topic, however, line 253-258 are missing two references.
19. The final conclusion of the paper states that „Based on our data it is speculated that amodiaquine may be the better artemisinin partner drug than piperaquine and lumefantrine on Bioko Island.“ I suggest that another sentence will be added calling for additional ex vivo susceptibility and clinical efficacy studies to adequately study this speculation.
20. The authors performed sequencing for Pfk13, Pfcrt and Pfmdr1. The sequencing data should be made available on the NCBI BankIT database.

Replies to Editor and Reviewers,

First of all, we thank both reviewers and editor for your positive and constructive comments and suggestions.

Then, we have read the comments carefully and made correction accordingly. In this letter, we have provided a response to each comment below. In the revised manuscript, all changes were highlighted. Furthermore, the Tables, Figures and supporting information were modified based on the **Microbiology Spectrum style**. We tried our best to improve the manuscript and made some changes in the manuscript. These changes will not influence the content and framework of the paper. And here we remain the changing trace in revised paper with revisions mode.

Finally, the manuscript was edited for proper English language, grammar, punctuation, spelling, and overall style by Grammarly Premium and AJE Digital Editing.

Once again, we appreciate for Editors/Reviewers' warm work earnestly, and hope that the correction will meet with approval. The detail reply information for the editors and reviewers is as follows.

With best regards

Yours sincerely

Jian Li, Min Lin

Specific responses to the reviewers's comments

Reviewer #1 (Comments for the Author):

Paper Summary

Lui et al describe the molecular markers of ACT resistance in a small convenience sample of approximately 100 patients with uncomplicated malaria at Malabo Regional Hospital from Jan 2017 - Dec 2019.

Major comments

Epi Methods

1. Samples collected over 2 years (2017-2019), this should be discussed as a limitation, as resistance could have changed over this time. Would recommend a sensitivity analysis to explore the effect of time. Break it up in years or 6-month buckets or another method. Is there a difference over time?

A: We strongly agree with the reviewer that the resistance could have changed over time. However, the time span and samples number of this study were limited to allow for correlation analysis. In addition, sample collection has been difficult due to the impact of the novel coronavirus epidemic. We will adopt this meaningful suggestion for an in-depth study when the number of subsequent samples reaches a certain number. Here we consider whether resistance and sensitivity analysis can be differentiated by the local rainy and dry seasons. We are grateful to the reviewers for this suggestion, and although we have not explored this time-limited effect in depth, this is an issue that we will focus on in subsequent studies, which will be an interesting topic for future research.

2. Lack of geographical representation. This should be discussed as a limitation. It appears that all samples came from Malabo Regional Hospital and Clinic of the Chinese Medical Aid Team. It is not clear if the results of this convenience sample can be generalized to the overall parasite population on the island

A: Our study was carried out in Malabo Regional Hospital, Bioko Island. Bioko, with an area of about 2,000 square kilometers, is an island 32 km off the west coast of Africa and its population is about 330,000 (2015 census) (PMID: 32660484). Because of its particular geographic posture of active volcanoes and tropical rainforests, about 90% of the population lives in the capital, Malabo, with only a small percentage living

in small villages outside of the city. Serious illnesses (including malaria) in these areas are generally cared for at the Malabo Regional Hospital, a government-owned public hospital (PMID: 34284778). Therefore, the sample we collected is somewhat geographically representative.

Our previous study analyzed the information underlying the genetic diversity of merozoite surface proteins 1 (MSP-1) and 2 (MSP-2) in *P. falciparum* from Malabo Regional Hospital, Bioko Island, Equatorial Guinea (PMID: 30526609). The results showed that the MAD20 (9 alleles) family dominated in *msp1*, followed by the K1 (9 alleles) and R033 (8 alleles) families. In *msp2*, the FC27 (5 alleles) family was the most frequently detected, followed by the 3D7 (20 alleles) family. It can be seen that although the geographical area of Bioko Island is small, mixed clones of *P. falciparum* are still present, which could explain the geographical limitations.

3. Results should be compared against a recent Therapeutic Efficacy Study (TES) that was not referenced. One of the three sites was Malabo. (PMID: 34158055). Do these molecular markers of resistance correlate with clinical efficacy from this recent TES?

A: Thank you for bringing issue to our attention. We agree and have included in the discussion a comparison of this study with the recent efficacy study of Malabo. It is specifically described as follows: In addition, another *Pfk13* mutation (*Pfk13_R561H*) associated with artemisinin resistance has been identified in Africa (PMID: 32747827). Bioko *Pfk13* resistance mutations were not observed in this study, and no significant selection effect was obtained by using bioinformatics tools. Previous studies have shown only three *Pfk13* mutations (*Pfk13_E433D* and *Pfk13_A578S*) were present in 476 samples after AS/AQ and AL treatment in the Equatorial Guinea region (including Bioko, Annobo'n and Corisco Bay) and were also not associated with known artemisinin resistance-related markers (Pmid: 34158055). These results demonstrated that artemisinin remains highly effective and supported the BIMEP recommendation above (**page 8, lines 238-245 in the revised version**).

Molecular Methods

1. Why were only monoclonal samples evaluated? What was done with the polyclonal sequences?

A: It is possible that our description was not very accurate, the concept we wanted to express was that the samples were only infected with *P. falciparum* and no other Plasmodium infections (determined by microscopy and a rapid diagnostic test (RDT),

confirmed by qPCR). The description "monoclonal" has been suitably modified in the revised version, e.g.: "monoclonal *Pfk13* whole sequences" (page 3, line 110 in the original version) has been revised to "*Pfk13* sequences were successfully amplified in the samples" (page 3, lines 113-114 in the revised version); "monoclonal infections" (page 7, line 198 and page 12, line 351 in the original version) has been revised to "infected cases" (page 6, line 205 and page 10, line 370 in the revised version).

2. How were monoclonal infections identified?

A: As with the above question, the results of this study by microscopy, RDT and qPCR showed that the samples involved only *P. falciparum* infections alone and no other *Plasmodium spp.* infections. Molecular investigations of the *Pfprt*, *Pfk13*, *Pfpm2* and *Pfmdr1* genes associated with *P. falciparum* were carried out for *P. falciparum* isolates.

3. Figure 1 - would be clearer presented as a table. Recommend including the gels (PFE) in a supplement.

A: The sequencing peak map and gel electrophoresis plot make the results more visual and explicit. We have bolded the mutation sites in Figure 1 so that the results can be displayed more prominently and clearly.

4. Figure 2 - Standard box plots would more clearly represent the data. The violin plots incorrectly show a tail of the distribution below the threshold in the left-most plot.

A: The reviewer is correct, and as suggested we have changed the violin plots in Figure 2 to box plots.

5. *Pfk13* Haplotype analysis: This analysis is very similar to a previously published report from 2019 on *Pfk13* diversity. PMID: 31085516. It appears to differ only by inclusion of the Bioko samples. Worryingly, almost entire sentence is copied and pasted verbatim from this manuscript (for example, lines 140-141 and lines 171-173). Notably, the earlier paper (PMID: 31085516) is not cited in this paper. Beyond the apparent plagiarism, it is not clear how much inclusion of Bioko samples adds to this previous analysis.

A: Thank you for pointing this mistake out. Indeed we cited the previous paper (PMID: 31085516) when describing the functions of dS and dN. We have redescribed dS and dN in the revised version and added this reference. The content has been modified as follows: Estimation of the number of synonymous substitutions per site (dS) and number of nonsynonymous substitutions per site (dN) explore the role of natural selection on *Pfk13* polymorphism (PMID: 31085516) (**page 4, lines 145-147 in the revised version**).

Previous studies have analyzed the genetic diversity of *Pfk13* several times, but none of them were related to Bioko Island. This study complemented the *Pfk13* sequences of Bioko for statistical analysis, not only to explore the relationship between *Pfk13* haplotypes from Bioko and other regions, but also to provide a complement to the *Pfk13* analysis in Africa. We also redescribed the TCS network of the *Pfk13* gene: The templeton crandall, and sing (TCS) network of the *Pfk13* gene is distributed using the two most dominant haplotypes, H_3 and H_10 radiation, which are shared by the parasites in Bioko, Africa, and Asia (**page 5, lines 179-181 in the revised version**).

Conclusions

1. Appropriateness of conclusions is overstated. It is not appropriate to recommend changing ACT therapy solely based molecular data from only approximately 100 samples without data on clinical efficacy, such as from a TES. The recent TES in Equatorial Guinea was not referenced and the results of this study were not interpreted in the context of this important source of efficacy data. Does the genotypic data from this study match the phenotypic data (recent TES)?

A: The reviewer's comments were excellent. To address this comment, we have substantially modified our manuscript. It is indeed inappropriate to speculate in the conclusion that amodiaquine may be the better artemisinin partner drug in Bioko based only on genetic mutations and multiple copies. It is indeed inappropriate to speculate in the conclusion that amodiaquine may be a better artemisinin partner drug in Bioko Island based only on genetic mutations and multiple copy numbers. In the revised manuscript we have removed the recommendation for artemisinin partner drugs from the conclusion and revised it as follows: Based on our data, it is recommended that continued monitoring of molecular markers of resistance is necessary in Bioko Island, which might provide baseline prevalence data to guide the use of ACTs. Furthermore, in conjunction with the results of molecular monitoring, additional ex vivo susceptibility and clinical efficacy studies are necessary to further

confirm the greater suitability of artemisinin partner drugs such as amodiaquine, piperazine and lumefantrine (**page 7, lines 287-291 in the revised version**).

We have added to the discussion a comparison of the *Pfk13* gene polymorphisms in this study with the Equatorial Guinea region (including Bioko). The results of these two studies are consistent, both showing no artemisinin resistance-associated mutations in *Pfk13*. This is described as follows: Previous studies have shown that only three *Pfk13* mutations (*Pfk13_E433D* and *Pfk13_A578S*) were present in 476 samples after AS/AQ and AL treatment in the Equatorial Guinea region (including Bioko, Annobo'n and Corisco Bay) and were also not associated with known artemisinin resistance-related markers (pmid: 34158055). These results demonstrated that artemisinin remains highly effective and supported the BIMEP recommendation above (**page 6, lines 241-245 in the revised version**).

Minor comments

1. Line 32: Prevalence of what? Mutations? Repeats?

A: Here is the prevalence of mutations, which has been modified to "the prevalence of *Pfk13*, *Pfprt*, and *Pfmdr1* mutations" (**pages 1-2 , lines 32-33 in the revised version**).

2. Line 38: "high efficacy" can only be determined with a TES.

A: The reviewer is correct. In the absence of TES, we cannot conclude that "artemisinin and its derivatives remains highly efficacy and amodiaquine considered as the preferential partner drug". We have modified this description to "The results suggested that the implementation of continuous drug monitoring in Bioko island is an essential measure" (**page 2, lines 38-39 in the revised version**).

3. Line 110: Describe why only 112 sequences were successfully obtained? Were they polyclonal?

A: Thank you for bringing issue to our attention. *Pfk13* gene amplification was performed on *P. falciparum*, and only 112 out of 152 samples succeeded in obtaining bands of corresponding size, so we consider that 112 *Pfk13* sequences were successfully obtained. In addition, although 112 *Pfk13* products were obtained by nested PCR and sequencing, some of the results were poor due to their long amplification sequences, and only 96 results could be used for subsequent analysis, which we have added in the manuscript: A total of 112 (73.68%, 112/152) *Pfk13*

sequences were successfully amplified in the samples, and only 96 of them were available for subsequent analysis due to their long sequences and poor partial sequencing results (**page 3, lines 113-115 in the revised version**).

4. Line 113: Cite references that show those mutations were not associated with artemisinin delayed clearance.

A: Thank you for bringing this issue to our attention. We have added references PMID: 27332904, PMID: 25075834 and PMID: 30651111 to the revised manuscript (**page 3, line 118 in the revised version**).

Details are as follows:

19. Ménard, D., Khim, N., Beghain, J., Adegnika, A. A., Shafiu-Alam, M., Amodu, O., Rahim-Awab, G., Barnadas, C., Berry, A., Boum, Y., Bustos, M. D., Cao, J., Chen, J. H., Collet, L., Cui, L., Thakur, G. D., Dieye, A., Djallé, D., Dorkenoo, M. A., Eboumbou-Moukoko, C. E., ... KARMA Consortium. 2016. A Worldwide Map of *Plasmodium falciparum* K13-Propeller Polymorphisms. *N Engl J Med.* 374:2453 – 2464.

20. Ashley, E. A., Dhorda, M., Fairhurst, R. M., Amaratunga, C., Lim, P., Suon, S., Sreng, S., Anderson, J. M., Mao, S., Sam, B., Sopha, C., Chuor, C. M., Nguon, C., Sovannaroeth, S., Pukrittayakamee, S., Jittamala, P., Chotivanich, K., Chutasmit, K., Suchatsoonthorn, C., Runcharoen, R., ... Tracking Resistance to Artemisinin Collaboration (TRAC). 2014. Spread of Artemisinin Resistance in *Plasmodium falciparum* Malaria. *N Engl J Med* 371:411 – 423.

21. WWARN K13 Genotype-Phenotype Study Group. 2019. Association of mutations in the *Plasmodium falciparum* Kelch13 gene (Pf3D7_1343700) with parasite clearance rates after artemisinin-based treatments — a WWARN individual patient data meta-analysis. *BMC Med* 17.

(**pages 12-13, lines 457-470 in the revised version**)

5. Line 136: Why were only 96/152 samples evaluated for *Pfk13*?

A: Although 112 *Pfk13* products were obtained by nested PCR and sequencing, some of the results were poor due to their long amplification sequences, and only 96 results could be used for subsequent analysis. We have added this point in the revised manuscript (**page 3, lines 113-115 in the revised version**).

6. Fig 3: Adding Bioko data to A, B, C would be helpful.

A: We've added the Bioko data to Figures 3A, B, and C in the revised version.

7. Line 171 and 182: Spell out acronyms at first use.

A: Done, The full name of TCS and EHH, templeton crandall, and sing (**page 5, line 179 in the revised version**) and extended haplotype homozygosity has been added, respectively (**page 5, lines 190-191 in the revised version**).

Meanwhile, all other abbreviations in the manuscript were checked and revised:

average number of pairwise nucleotide differences (K) (**page 4, line 141 in the revised version**);

haplotype diversity (Hd) (**page 4, lines 143-144 in the revised version**);

nucleotide diversity (π) (**page 4, line 144 in the revised version**);

number of synonymous substitutions per site (dS) (**page 4, line 145 in the revised version**);

number of nonsynonymous substitutions per site (dN) (**page 4, line 146 in the revised version**);

Linkage disequilibrium (LD) (**page 5, line 163 in the revised version**);

fixation index (F_{ST}) (**page 5, line 182 in the revised version**).

8. Line 218: Prior TES have been completed in Equatorial Guinea (including Malabo) and should be described and referenced. Results from TES and this study should be compared and contrasted.

A: The suggestion of reviewer was excellent and we have added this description to the revised manuscript as suggested: In 2017-2018, AS/AQ and AL have been shown to be highly effective in the treatment of uncomplicated *P. falciparum* infection in Equatorial Guinea (Bioko, Annobo'n and Corisco Bay) with cure rates of 98.6%-100% and 92.4%-100%, respectively. A small proportional mutation of *Pfk13* occurred which was not associated with artemisinin resistance. The data presented in this study are the first molecular investigations conducted for the *Pfcrt*, *Pfk13*, *Pfpm2* and *Pfmdr1* genes on Bioko island and assessed the polymorphic profile of the *Pfk13* gene. Accordingly, we aimed to provide some further useful references for assessing ACTs efficacy by monitoring molecular markers related to the drug resistance (**page 6, lines 225-232 in the revised version**).

9. Line 256: Leroy D? Is this a reference? If so, needs appropriate reference.

A: References were added to the revised version (**page 7, line 276 in the revised**

version):

34. Leroy D, Macintyre F, Adoke Y, Ouoba S, Barry A, Mombo-Ngoma G, Ndong NJM, Varo R, Dossou Y, Tshetu AK, Duong TT, Phuc BQ, Laurijssens B, Klopper R, Khim N, Legrand E, Menard D. 2019. African isolates show a high proportion of multiple copies of the *Plasmodium falciparum* plasmepsin-2 gene, a piperazine resistance marker. *Malar J* 18:126.

(page 14, lines 519-522 in the revised version)

Once again, we appreciate your warm work earnestly and hope that the correction will meet with approval. All that you mentioned for us will significantly improve the quality of our manuscript. We thank you again for your positive and constructive comments and suggestions.

Reviewer #2 (Comments for the Author):

The work by Liu et al. investigates the presence of mutations in PfKelch13, the marker associated with resistance to current artemisinin based antimalarials, and known resistance markers for partner drugs like amodiaquine and piperazine in a region in Bioko Island in Equatorial Guinea from 2017 to 2019. The findings indicate that there is little polymorphism in *Pfk13* on Bioko Island with one frequently observed mutation Pfk_K189T that is not associated with artemisinin resistance. Prevalence of *Pfprt* and *Pfmdr1* mutations associated with resistance to amodiaquine, piperazine and lumefantrine were fairly high indicating that artemisinin partner drug efficacy might be challenged on Bioko Island.

Current recommendations to treat malaria by the WHO are artemisinin combination therapies. Artemisinin resistance is prominent in Asia and first reports suggest association of specific PfKelch13 mutations with delayed parasite clearance in Africa. Surveillance of *Pfk13* mutations in African region is of great importance. Liu et al. report on surveillance of *Pfk13* mutations and partner drugs resistance markers *Pfprt* and *Pfmdr1* on Bioko Island in Equatorial Guinea. The results suggest little polymorphisms in *Pfk13*, but presence of resistance markers in *Pfmdr1* and *Pfprt*. The study adds to the understanding of circling *Pfk13* resistance markers in Africa and comparisons with other isolates from Africa and Asia highlight the separate emergence of *Pfk13* mutations in Africa and Asia.

Comments:

1. The manuscript is well written and conveys the messages clearly and concisely, however, there are several grammatical mistakes, words missing and abbreviations not explained. I suggest looking closely at the manuscript and fixing these issues. A few examples: Line 89, ACTs mentioned twice, line 94 should be Bioko Island reported

A: Thanks for the reviewer. The manuscript has been professionally edited by American Journal Experts (AJE) English editors.

We have corrected “ACTs ACTs” to “ACTs” (page 3, line 88 in the revised version); “On Bioko Island, it reported” to “Bioko Island reported” (page 3, line 93 in the revised version).

2. The abstract and importance section need to be overhauled in terms of grammatical, spelling errors and the usage of the wrong verbs and adjectives. Additionally, the grammatical errors make it difficult to understand which message the authors want to convey and can therefore, not be properly evaluated.

A: We have carefully checked the full text and as suggested the abstract and importance section has been redescribed in the revised version. The manuscript has also been professionally edited by American Journal Experts (AJE) English editors.

3. Line 32: The authors state that prevalence of *Pfk13*, *Pfcrt* and *Pfmdr1* was 73.68%, 78.29% and 75.66%. Since these are genes that are always present in *P. falciparum* (prevalence 100%) what are the numbers referring to? Are these numbers based on a specific mutation?

A: Thank you for pointing out these issues, which we misdescribed in the manuscript. The numbers indicated in the question above are the prevalence of mutations in the *Pfk13*, *Pfcrt* and *Pfmdr1* genes. We have marked in the revised manuscript: the prevalence of *Pfk13*, *Pfcrt*, and *Pfmdr1* mutations was 73.68%, 78.29%, and 75.66%, respectively (pages 1-2, lines 32-33 in the revised version). The prevalence of *Pfmdr1*_N86Y, *Pfmdr1*_Y184F, and *Pfcrt*_K76T mutations was 22.69%, 43.70%, and 26.95%, respectively. Five nonsynonymous mutations *Pfk13*_H136N, *Pfk13*_K189N, *Pfk13*_K189T, *Pfk13*_N194D and *Pfk13*_S213G were observed in the *Pfk13* gene.

4. Line 96: The manuscript states that the efficacy of DHA/PPQ has not been documented on Bioko Island. Is this also true for the other ACTs?

A: The efficacy of other ACTs AS/AQ and AL in Equatorial Guinea (Bioko, Annobo'n and Corisco Bay) is reported in a recent study (Pmid: 34158055). We have added a description of the relevant background and comparison with our results to the discussion of the revised manuscript. It is specifically described as follows: In 2017-2018, AS/AQ and AL have been shown to be highly effective in the treatment of uncomplicated *P. falciparum* infection in Equatorial Guinea (Bioko, Annobo'n and Corisco Bay) with cure rates of 98.6%-100% and 92.4%-100%, respectively (**page 6, lines 225-227 in the revised version**). Previous studies have shown only three *Pfk13* mutations (*Pfk13_E433D* and *Pfk13_A578S*) were present in 476 samples after AS/AQ and AL treatment in the Equatorial Guinea region (including Bioko) and were also not associated with known artemisinin resistance-related markers (Pmid: 34158055). These results demonstrated that artemisinin remains highly effective and supported the BIMEP recommendation above (**page 6, lines 240-245 in the revised version**).

5. Line 101: The last sentence is a conclusive statement that is not backed up by the data analyzed in the manuscript. To decide on treatment options for malaria patients, additional clinical efficacy studies of the drugs would need to be conducted. Consider revising this sentence to better reflect the outcome of this study.

A: We strongly agree with the reviewer that no absolute conclusions can be drawn in the absence of clinical efficacy studies. The sentence has been revised into “This might contribute to the subsequent treatment of malaria patients and malaria control.” (**page 3, lines 100-101 in the revised version**).

6. Line 104: The first paragraph could be combined with the following paragraph.

A: we have merged the above paragraph to the following paragraph as suggested (**page 3, lines 104-107 in the revised version**).

7. Line 112: The sentence states that none of the observed mutations was associated with delayed parasite clearance. Does this refer to previous studies or was this evaluated in this study? For the former, please revise sentence to reflect the reference to former studies and reference them, if the latter, how was this evaluated? Where IC50 data obtained for the isolates?

A: With reference to previous studies (PMID: 27332904, PMID: 25075834 and PMID: 30651111), the five *Pfk13* mutant loci we identified in the non-propeller region, all of which are not associated with delayed parasite clearance and artemisinin resistance. Relevant references have been added to the revised manuscript (**page 3, line 118 in the revised version**).

Details are as follows:

19 Ménard, D., Khim, N., Beghain, J., Adegnika, A. A., Shafiul-Alam, M., Amodu, O., Rahim-Awab, G., Barnadas, C., Berry, A., Boum, Y., Bustos, M. D., Cao, J., Chen, J. H., Collet, L., Cui, L., Thakur, G. D., Dieye, A., Djallé, D., Dorkenoo, M. A., Eboumbou-Moukoko, C. E., ... KARMA Consortium. 2016. A Worldwide Map of *Plasmodium falciparum* K13-Propeller Polymorphisms. *N Engl J Med.* 374:2453 – 2464.

20 Ashley, E. A., Dhorda, M., Fairhurst, R. M., Amaratunga, C., Lim, P., Suon, S., Sreng, S., Anderson, J. M., Mao, S., Sam, B., Sopha, C., Chuor, C. M., Nguon, C., Sovannaroeth, S., Pukrittayakamee, S., Jittamala, P., Chotivanich, K., Chutasmit, K., Suchatsoonthorn, C., Runchaoen, R., ... Tracking Resistance to Artemisinin Collaboration (TRAC). 2014. Spread of Artemisinin Resistance in *Plasmodium falciparum* Malaria. *N Engl J Med* 371:411 – 423.

21 WWARN K13 Genotype-Phenotype Study Group. 2019. Association of mutations in the *Plasmodium falciparum* Kelch13 gene (Pf3D7_1343700) with parasite clearance rates after artemisinin-based treatments — a WWARN individual patient data meta-analysis. *BMC Med* 17.

(pages 12-13, lines 457-470 in the revised version)

Also, we have added to the discussion the description of an existing reported *Pfk13* mutation (*Pfk13_R561H*) associated with artemisinin resistance (PMID: 32747827) **(page 6, lines 238-239 in the revised version)**.

The following references have been added:

27. Uwimana A, Legrand E, Stokes BH, Ndikumana J, Menard D. 2020. Emergence and clonal expansion of in vitro artemisinin-resistant *Plasmodium falciparum* kelch13 R561H mutant parasites in Rwanda. *Nat Med* 26. **(page 13, lines 488-490 in the revised version)**.

8. Line 116-117: The sentence talks about the *Pfmdr1* and *Pfcrt* mutations, but the reference to the figure is missing. Please indicate the correct figure reference here.

A: Thank you for pointing out these issues. We have added the relevant figure references in the revised version (**page 3, line 111 and line 113 in the revised version**).

9. Figure 1: The figure shows the mutations found in *Pfmdr1*, *Pfcrt* and *Pfk13*. The figure does not include the sequence change for *Pfcrt*_K76T. Please describe the schematic gene map and the colors/symbols in the figure legend. C is not mentioned in the text and is not informative for the reader and should be removed from the main figure. Additionally, electropherograms for *Pfk13*_N194D and *Pfk13*_S213G are of poor quality and actually showing mixed infection for these two mutations. This should be reflected in the text and analysis.

A: We have redescribed *Pfcrt*_K76T in the FIG 1, and the legend (including gene map and colors/symbols): Green, red, black and blue peaks indicate A, T, G, C bases, respectively. Mutated bases or amino acids are marked in bold. Underlines indicate linked mutations (**page 4, lines 123-124 in the revised version**).

FIG 1C is mentioned in the method "PCR amplification and sequencing of *Pfmdr1*, *Pfcrt*, and *Pfk13*" (**page 9, line 336 in the revised version**). Considering that the electrophoresis gel images is more visual, we put it in the main figure.

We rescaled the resolution of the *Pfk13*_N194D and *Pfk13*_S213G electropherograms. Additional descriptions have been added for mixed infections of these two mutations: Using PF3D7_1343700 as the reference sequence, five nonsynonymous mutations (H136N, K189N, K189T, VN193-194V/AD, S213G/A/T) were observed, but none of them was associated with artemisinin delayed parasite clearance (19, 20, 21). Among them, *Pfk13*_K189T (53.12%) was the most prevalent in the Bioko isolates, while the other loci were mutated at a lower frequency. The mutations at codons 193-194 and 213 were mixed type, with amino acids mutated from VN to V/AD and S to G/A/T, respectively (**pages 3-4, lines 115-121 in the revised version**).

10. Line 126: The authors state that they used a threshold of 1.5 to define single or multiple copies of *Pfpm2*. How was this threshold determines? It would be more accurate to have a "positive" control with 2 copies of *Pfpm2*. I would suggest running

a few additional qPCRs with a 2 copy and a 1 copy control to precisely determine the threshold.

A: We used a threshold of 1.5 to define single or multiple copies of *Pfpm2* by referring to the reference PMID: 30967148, which suggested that multiple copies vs single copy of *Pfpm2*, were defined as copy numbers < 1.5 and ≥ 1.5 , respectively. We strongly agree with the reviewer's suggestion of a more accurate "positive" control of 2 *Pfpm2* copies and a 2-copy and 1-copy control to precisely determine the threshold. We will further adopt the implementation of this recommendation in the subsequent study.

11. Line 129-130: The authors speculate that due to increase in *Pfpm2* copy number variation drug resistance has emerged in Bioko Island. This is a bold statement that is only made on *Pfpm2* copy number variation, experimental susceptibility assays or clinical data is missing. Therefore, I suggest rephrasing this sentence and using more cautious words to describe the potential for drug resistance. Additionally, it would be good to state which drug would be effected.

A: Thank you for pointing this issue out. Indeed, it is inappropriate to describe directly the resistance of Bioko parasites if only statistics of *Pfpm2* copy number variation are available, without experimental susceptibility assays and clinical data. As suggested we modified the sentence as follows: Based on the increased CNVs of the *Pfpm2* gene on Bioko Island compared to the standard strain 3D7 of *P. falciparum*, it is hypothesized that the *Pfpm2* CNVs may serve as a risk assessment indicator for tracking the development and spread of DHA/PPQ drug resistance on the island (**page 4, lines 132-135 in the revised version**).

12. The manuscript uses many different biostatistical analyses for the relatedness and genetic diversity of the isolates. It is important to clarify all the abbreviations used to describe these analyses. Examples are: Line 136: "K value", line 139 "Hd" + " π ", Line 141 "dS" and "dN", line 157 "LD", Line 174 "FST index", Line 198 "TCS", Line 204 "FST", Line 182 "EHH"

A: Thank you for pointing out these issues. We have added specific descriptions of these abbreviations:
average number of pairwise nucleotide differences (K) (**page 4, line 141 in the revised version**);
haplotype diversity (Hd) (**page 4, lines 143-144 in the revised version**);

nucleotide diversity (π) (page 4, line 144 in the revised version);
number of synonymous substitutions per site (dS) (page 4, line 145 in the revised version);
number of nonsynonymous substitutions per site (dN) (page 4, line 146 in the revised version);
Linkage disequilibrium (LD) (page 5, line 163 in the revised version);
fixation index (F_{ST}) (page 5, line 182 in the revised version);
templeton crandall, and sing (TCS) (page 5, line 179 in the revised version);
extended haplotype homozygosity (EHH) (page 5, lines 190-191 in the revised version).

13. Figure 3: Figure 3 shows the mutation frequency in *Pfk13* in African and Asian parasites. Figure 3B: It would be good to describe the frequency scale in the figure legend as there are two different parameters (color and size of square) listed for the mutation frequency.

A: We have added the description of the frequency scale to the legend of Figure 3B: Mutation frequency of *Pfk13* gene in different regions. The value is indicated by the square size and color: the larger the square and the redder the color, the higher the mutation frequency (page 5, lines 172-174 in the revised version).

14. Line 210: The authors state that there is a high rate of malaria transmission on Bioko Island, however, in the introduction it is stated that the transmission rate went down from 45% to 8.5%. Can you elaborate on this?

A: Many previous studies have pointed out that malaria is a major health problem on Bioko Island, with historically high transmission rates (PMID: 34284778 and PMID: 29402288). Since 2004, through the Bioko Island Malaria Control Project (BIMCP), the island-wide implementation of measures such as the use of indoor residual spraying and long-lasting insecticides, the introduction of ACTs and comprehensive surveillance, which had reduced the parasite prevalence on the island from 43.3% to 8.5% (PMID: 29402288 and <https://mcdinternational.org/bimcp>). Thus, although malaria transmission on Bioko has been significantly reduced through various interventions, the island remains endemic compared to other areas and the local population is still at high risk of infection.

15. Line 218: The authors state that the study is looking at the efficacy of ACTs by looking at molecular markers. To determine efficacy of the drugs the study would have needed to include either susceptibility data of isolates or clinical efficacy data. I advice to revise this sentence to adequately reflect the topic of the paper e.g. "The data presented here describe the first study on Bioko Island looking at molecular markers associated with ACT drug resistance".

A: We have adopted the reviewers' comments and revised the sentence as follows: The data presented in this study are the first molecular investigations conducted for the *Pfcr1*, *Pfk13*, *Pfpm2* and *Pfmdr1* genes on Bioko island and evaluated polymorphic profile of the *Pfk13* gene. Accordingly, we aimed to provide some further useful references for assessing ACTs efficacy by monitoring molecular markers related to the drug resistance (**page 6, lines 228-232 in the revised version**).

16. Line 233-236: The conclusion for the *Pfk13*_K189T found in Bioko parasites is that it does not influence artemisinin resistance; but that it might be good to study this mutation further to understand why artemisinin resistance is developing more slowly in Africa. How does this mutation have an influence on artemisinin resistance development? Can you elaborate further on this speculation?

A: *Pfk13*_K189T haplotype analysis revealed a clear evolutionary divergence between *P. falciparum* in Southeast Asia and Africa. At the same time, the *Pfk13* mutant loci in these two regions were significantly different, with the locus in Southeast Asia effectively indicating artemisinin resistance in the local strain, but the resistance locus in the African strain did not effectively indicate the corresponding resistance. It is suggested that artemisinin resistance markers may differ between these two regions and that there are unexplored genes involved in African artemisinin resistance.

The current level of possible artemisinin resistance in Africa is not low, it is just that the available resistance loci are not effectively indicated. Therefore, subsequent new genes and loci for artemisinin resistance need to be re-excavated. This will also be our next key research direction.

17. Line 249: The authors state that the *Pfmdr1* ad *Pfcr1* mutation frequency decreased compared to previous findings. However, the authors also noted that this is the first study on Bioko Island. Can you clarify this point?

A: Although we have done some previous studies on *Pfmdr1* and *Pfcr1* in Bioko Island (PMID: 26325683, PMID: 25348116), our study is the first systematic molecular monitoring of the *Pfcr1*, *Pfk13*, *Pfpm2* and *Pfmdr1* genes in Bioko Island. Compared with the previous studies, the data in this study are more systematic and comprehensive, which can better provide a reference for the assessment of drug resistance in ATCs.

18. References are overall appropriate for the topic, however, line 253-258 are missing two references.

A: We would like to thank the reviewer for this suggestion. The followed two references were added to the revised manuscript:

33. Imwong M, Hien TT, Thuy-Nhien NT, Dondorp AM, White NJ. 2017. Spread of a single multidrug resistant malaria parasite lineage (*PfPailin*) to Vietnam. *Lancet Infect Dis* 17:1022.

34. Leroy D, Macintyre F, Adoke Y, Ouoba S, Barry A, Mombo-Ngoma G, Ndong NJM, Varo R, Dossou Y, Tshetu AK, Duong TT, Phuc BQ, Laurijssens B, Klopper R, Khim N, Legrand E, Menard D. 2019. African isolates show a high proportion of multiple copies of the *Plasmodium falciparum* plasmepsin-2 gene, a piperazine resistance marker. *Malar J* 18:126.

(page 14, lines 517-522 in the revised version)

19. The final conclusion of the paper states that „ Based on our data it is speculated that amodiaquine may be the better artemisinin partner drug than piperazine and lumefantrine on Bioko Island." I suggest that another sentence will be added calling for additional ex vivo susceptibility and clinical efficacy studies to adequately study this speculation.

A: We have modified this sentence as suggested: Based on our data, it is recommended that continued monitoring of molecular markers of resistance is necessary in Bioko Island, which might provide baseline prevalence data to guide the use of ACTs. Furthermore, in conjunction with the results of molecular monitoring, additional ex vivo susceptibility and clinical efficacy studies are necessary to further confirm the greater suitability of artemisinin partner drugs such as amodiaquine, piperazine and lumefantrine **(page 7, lines 287-291 in the revised version)**.

20. The authors performed sequencing for *Pfk13*, *Pfcrt* and *Pfmdr1*. The sequencing data should be made available on the NCBI BankIT database.

A: Based on this suggestion we will subsequently submit and refine the sequencing data of *Pfk13*, *Pfcrt* and *Pfmdr1* in the NCBI BankIt database.

Once again, we appreciate your warm work earnestly and hope that the correction will meet with approval. All that you mentioned for us will significantly improve the quality of our manuscript. We thank you again for your positive and constructive comments and suggestions.

April 20, 2022

Dr. Min Lin
Hanshan Normal University
School of Food Engineering and Biotechnology
chaozhou
China

Re: Spectrum00413-22R1 (Molecular Surveillance of Artemisinin-based Combination Therapies Resistance in *Plasmodium falciparum* Parasites from Bioko Island, Equatorial Guinea)

Dear Dr. Min Lin:

I have reviewed your resubmission and the reviewer comments. Please respond to this second review and make the recommended changes as this manuscript is not, at present, ready for publication.

Thank you for submitting your manuscript to Microbiology Spectrum. As you will see your paper is very close to acceptance. Please modify the manuscript along the lines I have recommended. As these revisions are quite minor, I expect that you should be able to turn in the revised paper in less than 30 days, if not sooner. If your manuscript was reviewed, you will find the reviewers' comments below.

When submitting the revised version of your paper, please provide (1) point-by-point responses to the issues I raised in your cover letter, and (2) a PDF file that indicates the changes from the original submission (by highlighting or underlining the changes) as file type "Marked Up Manuscript - For Review Only". Please use this link to submit your revised manuscript. Detailed instructions on submitting your revised paper are below.

Link Not Available

Sincerely,

Laura Kirkman

Reviewer comments:

Reviewer #1 (Comments for the Author):

Paper Summary

Lui et al describe the molecular markers of ACT resistance in a small convenience sample of approximately 100 patients with uncomplicated malaria at Malabo Regional Hospital from Jan 2017 - Dec 2019.

The authors have thoroughly responded to the major and minor comments in the previous review, however it appears some recommended changes and concepts were not adjusted in the manuscript.

Please see additional Major and Minor comments below.

Major comments

1. From the first review, it appears that the authors did not change the manuscript to include the multiple limitations to this study.

The duration of time sample collection (2017-2019) where resistance patterns likely changed, lack of geographical representation etc. These should be described as limitations in the discussion section.

2. Figure 1 - The sequencing peak map can be simplified to table or even few sentences describing the mutations. Also, the gel electrophoresis image does not add much to the manuscript as the PCR products are shown in Table 1.

3. Abstract Line 28: states "to evaluate the effect" this study does not evaluate effects of first line ACTs. It is only looking at molecular markers of resistance. The authors continue to confuse the idea of effect or efficacy from a Therapeutic Efficacy Study (TES) with surveillance of molecular markers of resistance. Genotypic markers of resistance do not always correlate with phenotypic resistance to ACTs in clinical practice. The authors need to describe and make clear the differences between surveillance of genetic markers compared to true clinical efficacy in a TES.

Minor Comments

1. Line 23: The first sentence of the abstract is quite bold. "Artemisinin-based combination therapies (ACTs) resistance has emerged and diffused in Africa." It is true that resistance has emerged, however many TES results and even data here suggest that it has not year diffused in Africa. I would remove the word diffused or state that it "could be diffusing" in Africa.

2. Line 97: "to evaluate the effect" again same argument. This study is not evaluating the effect of a medication. This is not a TES.

3. Line 290: "Clinical efficacy" should be changed to "ongoing therapeutic efficacy studies" (TES). TES should be abbreviated when used elsewhere in the manuscript.

4. Line 230-231: "Accordingly, we aimed to provide some further useful references for assessing ACTs efficacy by monitoring molecular markers related to the drug resistance." Does this mean you are adding references of previously completed TESs? Or are you again stating this study is trying to measure efficacy of ACTs? This is unclear. Again, this study is only looking at genotypic markers of resistance, not efficacy of ACTs.

Preparing Revision Guidelines

- point-by-point responses to the issues I raised in your cover letter
- Upload a compare copy of the manuscript (without figures) as a "Marked-Up Manuscript" file.
- Each figure must be uploaded as a separate file, and any multipanel figures must be assembled into one file.
- Manuscript: A .DOC version of the revised manuscript
- Figures: Editable, high-resolution, individual figure files are required at revision, TIFF or EPS files are preferred

Please return the manuscript within 60 days; if you cannot complete the modification within this time period, please contact me. If you do not wish to modify the manuscript and prefer to submit it to another journal, please notify me of your decision immediately so that the manuscript may be formally withdrawn from consideration by Microbiology Spectrum.

Replies to Editor and Reviewers,

We thank both reviewers and editor for your positive and constructive comments and suggestions on our manuscript (Molecular Surveillance of Artemisinin-based Combination Therapies Resistance in *Plasmodium falciparum* Parasites from Bioko Island, Equatorial Guinea, ID: Spectrum00413-22R1), and we gratefully appreciate the chance to revise our manuscript. We would like to sincerely thank the reviewers for their valuable suggestions, which helped us significantly improve our manuscript.

We have carefully considered all the comments provided and revised our manuscript accordingly. These changes will not influence the content and framework of the article. A detailed, point-by-point response to the comments and the revised text are enclosed. The revised version has been approved by all the coauthors.

The manuscript was edited for proper English language, grammar, punctuation, spelling, and overall style by AJE Digital Editing.

Once again, we appreciate for Editors/Reviewers' warm work earnestly, and hope that the correction will meet with approval. The detail reply information for the editors and reviewers is as follows.

With best regards

Yours sincerely

Jian Li, Min Lin

Specific responses to the reviewers's comments

Reviewer #1 (Comments for the Author):

Paper Summary

Lui et al describe the molecular markers of ACT resistance in a small convenience sample of approximately 100 patients with uncomplicated malaria at Malabo Regional Hospital from Jan 2017 - Dec 2019.

The authors have thoroughly responded to the major and minor comments in the previous review, however it appears some recommended changes and concepts were not adjusted in the manuscript.

Please see additional Major and Minor comments below.

Major comments

1. From the first review, it appears that the authors did not change the manuscript to include the multiple limitations to this study. The duration of time sample collection (2017-2019) where resistance patterns likely changed, lack of geographical representation etc. These should be described as limitations in the discussion section.

A: Thank you for pointing out these issues, which was an oversight on our part. The limiting effects of factors such as time, geography and sample number have already been added to the discussion. It is specifically described as follows:

Sample collection for this study was carried out in Malabo Regional Hospital, Bioko Island. Bioko, with an area of approximately 2,000 square kilometers, is an island 32 km off the west coast of Africa and its population is approximately 330,000 (2015 census) (25). Because of its particular geographic posture of active volcanoes and tropical rainforest, approximately 90% of the population lives in the capital, Malabo, with only a small percentage living in small villages outside of the city. Serious illnesses (including malaria) in these areas are generally cared for at the Malabo Regional Hospital, a government-owned public hospital (26). Therefore, the sample we collected is somewhat geographically representative. A previous study analyzed the genetic diversity of merozoite surface proteins 1 (MSP-1) and 2 (MSP-2) in *P.*

falciparum from Malabo Regional Hospital. The results showed that the MAD20 (9 alleles) family dominated in *msp1*, followed by the K1 (9 alleles) and R033 (8 alleles) families. In *msp2*, the FC27 (5 alleles) family was the most frequently detected, followed by the 3D7 (20 alleles) family (27). Although the geographical area of Bioko Island is small, mixed clones of *P. falciparum* are still present, which further indicates that the *Plasmodium* parasites in the region are geographically representative. Unfortunately, the limited time span and number of samples collected, coupled with the increased difficulty of sample collection due to the novel coronavirus epidemic. Therefore, further exploration between time and resistance is hampered, which will be the focus of our later studies (**page 7, lines 231-247**).

The references cited and added were:

25. Huang H-Y, Liang X-Y, Lin L-Y, Chen J-T, Ehapo CS, Eyi UM, Li J, Jiang T-T, Zheng Y-Z, Zha G-C, Xie D-D, He J-Q, Chen W-Z, Liu X-Z, Mo H-T, Chen X-Y, Lin M. 2020. Genetic polymorphism of *Plasmodium falciparum* circumsporozoite protein on Bioko Island, Equatorial Guinea and global comparative analysis. *Malar J* 19:245.

26. Nchama VUNN, Said AH, Mtoro A, Bidjimi GO, Owono MA, Maye ERM, Mangue MEO, Okomo GNN, Pasialo BEN, Ondo DM, Lopez M-SA, Mochomuemue FL, Obono MO, Besaha JCM, Chuquiyauri R, Jongo SA, Kamaka K, Kibondo UA, Athuman T, Falla CC, Eyono JNM, Smith JM, García GA, Raso J, Nyakarungu E, Mpina M, Schindler T, Daubenberger C, Lemiale L, Billingsley PF, Sim BKL, Richie TL, Church LWP, Olotu A, Tanner M, Hoffman SL, Abdulla S. 2021. Incidence of *Plasmodium falciparum* malaria infection in 6-month to 45-year-olds on selected areas of Bioko Island, Equatorial Guinea. *Malar J* 20:322.

27. Chen JT, Li J, Zha GC, Huang G, Huang ZX, Xie DD, Zhou X, Mo HT, Eyi JUM, Matesa RA, Obono MMO, Li S, Liu XZ, Lin M. 2018. Genetic diversity and allele frequencies of *Plasmodium falciparum msp1* and *msp2* in parasite isolates from Bioko Island, Equatorial Guinea. *Malar J* 17:458 (**page 14, lines 497-511**).

Subsequent references were renumbered in order.

2. Figure 1 - The sequencing peak map can be simplified to table or even few sentences describing the mutations. Also, the gel electrophoresis image does not add much to the manuscript as the PCR products are shown in Table 1.

A: In accordance with the reviewer's comments, we have replaced Figure 1 with Table 1. The corresponding statements in the text regarding "Figure 1" have been changed

to “Table 1” (**page 4, lines 121-124**). Also, the subsequent figure notes “Figures 2, 3, and 4” have been changed to Figures “1 (**page 4, line 134**), 2 (**page 5, line 169**), and 3 (**page 6, line 203**)”, respectively, and “Table 1” was changed to “Table 2” (**page 9, line 357**). We have highlighted all changes in the revised version.

3. Abstract Line 28: states "to evaluate the effect" this study does not evaluate effects of first line ACTs. It is only looking at molecular markers of resistance. The authors continue to confuse the idea of effect or efficacy from a Therapeutic Efficacy Study (TES) with surveillance of molecular markers of resistance. Genotypic markers of resistance do not always correlate with phenotypic resistance to ACTs in clinical practice. The authors need to describe and make clear the differences between surveillance of genetic markers compared to true clinical efficacy in a TES.

A: The reviewer is correct. we have described the molecular markers of drug resistance in the manuscript (**pages 2-3, lines 70-80**). It is indeed that monitoring of genetic markers does not indicate the effectiveness of clinical drugs. We have rephrased the sentence in the revised version as follows: Molecular monitoring targeting the *Pfcr1*, *Pfk13*, *Pfpm2* and *Pfmdr1* genes was conducted to provide insight into the impact of current antimalarial drug resistance on the island (**page 1, lines 26-28**). We also carefully checked the whole text, and with subsequent comment alerts, ensured that no similar incorrect statements were made.

Minor Comments

1. Line 23: The first sentence of the abstract is quite bold. "Artemisinin-based combination therapies (ACTs) resistance has emerged and diffused in Africa." It is true that resistance has emerged, however many TES results and even data here suggest that it has not year diffused in Africa. I would remove the word diffused or state that it "could be diffusing" in Africa.

A: We thank the reviewer for pointing out this issue and for providing us with an appropriate description. We have revised “diffuse” to “could be diffusing” as suggested (**page 1, line 23**).

2. Line 97: "to evaluate the effect" again same argument. This study is not evaluating the effect of a medication. This is not a TES.

A: Here again, we have made a mistake in the description, we have modified “to evaluate the effect” to “to evaluate the resistance of *P. falciparum* to antimalarial drugs” (**page 3, line 97**).

3. Line 290: "Clinical efficacy" should be changed to "ongoing therapeutic efficacy studies" (TES). TES should be abbreviated when used elsewhere in the manuscript.

A: We have incorporated the suggestions and made changes. Because “therapeutic efficacy studies” appears elsewhere (**page 6, line 219**) in the manuscript in front of here, we have marked “therapeutic efficacy studies (TES)” in front (**page 6, line 219**), and changed here to “ongoing TES” (**page 8, line 305**). At the same time, we checked and verified the full text.

4. Line 230-231: "Accordingly, we aimed to provide some further useful references for assessing ACTs efficacy by monitoring molecular markers related to the drug resistance." Does this mean you are adding references of previously completed TESs? Or are you again stating this study is trying to measure efficacy of ACTs? This is unclear. Again, this study is only looking at genotypic markers of resistance, not efficacy of ACTs.

A: Here we still make the same descriptive error. Monitoring of molecular markers can only provide a reference for resistance to ACT partner drugs, not prove the efficacy of ACTs. The sentence has been modified to: Accordingly, we aimed to provide some useful references for the assessment of resistance to ACT partner drugs by monitoring molecular markers associated with the drug resistance (**page 7, lines 228-230**).

Once again, we appreciate your warm work earnestly and hope that the correction will meet with approval. All that you mentioned for us will significantly improve the quality of our manuscript. We thank you again for your positive and constructive comments and suggestions.

May 10, 2022

Dr. Min Lin
Hanshan Normal University
School of Food Engineering and Biotechnology
chaozhou
China

Re: Spectrum00413-22R2 (Molecular Surveillance of Artemisinin-based Combination Therapies Resistance in *Plasmodium falciparum* Parasites from Bioko Island, Equatorial Guinea)

Dear Dr. Min Lin:

Your manuscript has been accepted, and I am forwarding it to the ASM Journals Department for publication. You will be notified when your proofs are ready to be viewed.

Sincerely,

Laura Kirkman
Editor, Microbiology Spectrum
